# Stabilizing Equation Learning via Zero-Point Constraints

**Sannyuya Liu**[1 2]  **Ao Chen**[2 3]  **Lin Liu**[4]  **Ruxia Liang**[5]  **Xiaoxuan Shen**[1 2 *]  **Jianwen Sun**[1 2 3 *]

## Abstract

Symbolic Regression aims to discover interpretable mathematical expressions from data. Equation Learner (EQL) is a gradient-based method with strong fitting capability and expressive potential, yet it often activates redundant operators as model complexity grows, leading to over-complex expressions and unstable equation recovery. We analyze a gradient residual issue induced by operators that do not vanish at zero, which can prevent the ideal sparse expression from acting as a stable attractor during training and bias training toward unnecessarily complex structures, making exact recovery highly unreliable in practice. To address this, we propose EQL-Z, a structurally controllable symbolic regression framework. EQL-Z enforces zero-point constraints via zero-point consistent operator transformations to eliminate residual gradients on silent paths, and performs a small-to-large structure search that grows depth/width from a compact seed under a complexity-penalized validation score. After selecting a compact structure, we apply BFGS fine-tuning to refine coefficients. Experiments on synthetic and real-world datasets show that EQL-Z substantially improves exact equation recovery and in-/out-of-distribution generalization over vanilla EQL, achieving performance close to leading symbolic regression baselines. Code is available at https://github.com/Caaaa-a/EQL-Z.

*Corresponding authors. [1]Laboratory for Artificial Intelligence and New Forms of Education, Central China Normal University, Wuhan, Hubei, China [2]Faculty of Artificial Intelligence in Education, Central China Normal University, Wuhan, Hubei, China [3]National Engineering Research Center for Educational Big Data, Central China Normal University, Wuhan, Hubei, China [4]School of Law, Humanities and Sociology, Wuhan University of Technology, Wuhan, Hubei, China [5]School of Computer Science, Central China Normal University, Wuhan, Hubei, China. Correspondence to: Xiaoxuan Shen <shenxiaoxuan@ccnu.edu.cn>, Jianwen Sun <sunjw@ccnu.edu.cn>.

*Proceedings of the 43rd International Conference on Machine Learning*, Seoul, South Korea. PMLR 306, 2026. Copyright 2026 by the author(s).

## 1. Introduction

Symbolic regression (SR) aims to recover structurally compact and interpretable mathematical expressions from data, and plays an important role in physical modeling and scientific discovery (Cornelio et al., 2023; Deng et al., 2025; Li et al., 2024; Reinbold et al., 2021; Udrescu & Tegmark, 2020). In recent years, neural networks have been introduced into this field, giving rise to the line of research on neural symbolic regression. Equation Learner (EQL) (Martius & Lampert, 2016) is a representative method in this paradigm: it integrates differentiable symbolic operators (e.g., sin, · · ·, exp) into neural networks and, together with sparse regularization, guides the model to learn analytic expressions from data. EQL provides a flexible differentiable parameterization of symbolic expressions and has shown strong potential for extrapolative function learning. However, when the candidate network is large or over-parameterized, EQL often activates redundant operators, leading to unnecessarily complex expressions and unstable exact equation recovery.

To better understand one important source of this redundancy problem, we design a simple diagnostic task with a unique analytic solution and observe that EQL gradually drifts away from the ground-truth structure during training, with redundant terms becoming activated. A further gradient analysis shows that when certain operators do not vanish at zero, they can induce non-zero gradients even when their incoming connection weights are zero, causing these operators to be spuriously activated during optimization. Consequently, even if the target expression is representable by the network, the corresponding sparse parameter configuration may fail to behave as a stable training attractor before the prediction residual vanishes.

To address this residual-gradient source of instability, we propose EQL-Z, a structurally controllable training framework for symbolic neural networks. EQL-Z tackles redundant-activation failures from two complementary perspectives: it reduces the number of redundant paths exposed to optimization, and it mitigates the residual-gradient effect of redundant paths that remain inside a candidate structure. These two perspectives are instantiated by two key components:

1. **Small-to-large structure search.** Starting from a compact seed architecture, EQL-Z iteratively expands the network through local width/depth modifications and accepts a new structure according to a validation score penalized by symbolic complexity. Compared with training a single large, fixed network, this incremental strategy reduces unnecessary operator paths and lowers the risk of converging to over-complex local optima.

2. **Zero-point constraint.** EQL-Z uses zero-point-consistent transformations so that symbolic operators satisfy $\Lambda(0) = 0$, ensuring that a redundant path with zero connection weight and zero input to the operator produces zero output and does not receive residual gradients caused solely by a non-zero operator response at the origin.

These two components are complementary: the small-to-large search reduces redundant structures at the architectural level, while the zero-point constraint stabilizes the remaining redundant paths at the optimization level. Together, they preserve the interpretability of EQL while substantially improving exact formula recovery, leading to significant gains in both in-distribution and out-of-distribution generalization over the original EQL.

## 2. Related Work

Symbolic regression (SR) aims to recover structurally compact and interpretable mathematical expressions from data, and has been widely applied to physical modeling and scientific discovery (Matsubara et al., 2022; Makke & Chawla, 2024; Bendinelli et al., 2023; Udrescu et al., 2020). Classical approaches primarily rely on explicit search over symbolic structures and can be broadly divided into evolutionary and reinforcement learning (RL) based methods. Evolutionary methods are typified by genetic programming (GP) (Topchy et al., 2001; Kommenda et al., 2020), which simulates natural evolution to generate candidate expressions, with common extensions such as GOMEA (Virgolin et al., 2021) and semantic backpropagation GP (SBP-GP) (Pawlak et al., 2014). RL-based methods instead formulate expression construction as a state-action process with reward-driven search, as exemplified by DSR (Petersen et al., 2019) and NGGP (Mundhenk et al., 2021). Despite their distinct design philosophies for structure search, both families typically require substantial heuristic tuning and suffer from low search efficiency, making it difficult to meet the demands of efficient expression modeling in complex tasks.

To overcome these limitations, recent work has introduced neural networks into SR, giving rise to two main lines of research: transformer-based generative symbolic regression and neural networks for symbolic regression (Li et al., 2023;

Kubalík et al., 2023). The former leverages language models to directly generate expression structures that satisfy syntactic constraints, with representative work based on Transformer-style architectures (Li et al., 2022b; Shojaee et al., 2023; Biggio et al., 2021; Kamienny et al., 2022; Becker et al., 2023). However, their generative behavior is tightly tied to pretraining and fixed decoding strategies, lacking dynamic adaptation to data difficulty and struggling to construct complex expressions in a gradual, data-driven manner.

In contrast, neural networks for symbolic regression, exemplified by Equation Learner (EQL), embed differentiable symbolic operators (e.g., addition, multiplication, trigonometric functions) directly into neural architectures. This paradigm combines the efficient optimization of neural networks with the interpretability of symbolic models, enabling natural expression compression and structural sparsity control during training, and thus offering stronger control over the search process. Subsequent variants further enhance the expressiveness and structural controllability of this family: EQL÷ (Sahoo et al., 2018) introduces differentiable division and extrapolation-based validation to improve function modeling; iEQL (Werner et al., 2021) incorporates $L_0$ regularization and prior knowledge to guide more controllable structures and support richer operators (e.g., logarithms and division); OccamNet (Dugan et al., 2020) and MathONet (Zhou & Pan, 2022) reinforce sparsity in connections to increase flexibility; and CONSOLE (Li et al., 2022a) as well as DYSYMNET (Wu et al., 2024) focus on improving structural reachability and convergence efficiency.

Despite the strong representational power and fitting performance of EQL and its derivatives, overly complex expressions and unstable training can still arise in practice. Prior studies have reported gradient instabilities and a tendency to get trapped in poor local minima in the EQL family, which lead to unnecessarily complex formulas and degraded interpretability (Petersen et al., 2019; Werner et al., 2021). We further observe that even when the target function is exactly representable by the current architecture, the optimization trajectory can drift away from the ground-truth structure, indicating a lack of structural stability. One contributing factor is that existing work primarily emphasizes connection sparsity (e.g., pruning strategies), while largely overlooking how activation sparsity and gradient propagation jointly affect the controllability of symbolic expressions. In conventional neural networks, activation sparsity is often induced by nonlinearities such as ReLU (Nair & Hinton, 2010) and can be further reinforced by regularization or structural optimization (Cheng et al., 2017; Hoefler et al., 2021; Loni et al., 2023). In symbolic neural networks, however, activation sparsity is not only about computational efficiency, but directly affects whether the learned expression can stably converge to a compact structure. In particular, within

the EQL framework, both structural sparsity and gradient sparsity are useful for suppressing redundant terms and maintaining compact, interpretable expressions.

To address this source of instability in structure recovery and redundant activations in current EQL models, we propose *EQL-Z*, a new training mechanism that enhances structural control from both the operator-design and training-dynamics perspectives, with the dual goals of expression simplification and model compression.

## 3. Empirical Observations of Optimization Instability

We begin by formalizing the basic structure of the EQL network and its symbolic computation pipeline. We then analyze a concrete case study of fitting a target formula, focusing on the network's gradient response and its ability to recover the ideal analytic structure.

### 3.1. EQL Network Structure Definition

Let the input sample be $x \in \mathbb{R}^d$, where $d$ denotes the dimensionality of the input variables. An EQL network consists of $L$ layers, where layer $0$ is the input layer, layer $L$ is the output layer, and the remaining layers are hidden layers; see Figure 4 in Appendix A.

In each hidden layer, the network first applies a linear transformation to its input. The linear output of layer $l$ is denoted by $z^{(l)} \in \mathbb{R}^{m_l}$ and is computed as

$$z^{(l)} = W^{(l)} a^{(l-1)} + b^{(l)},$$

where $W^{(l)} \in \mathbb{R}^{m_l \times n_{l-1}}$ is the weight matrix of layer $l$, $b^{(l)} \in \mathbb{R}^{m_l}$ is the bias vector, and $a^{(l-1)} \in \mathbb{R}^{n_{l-1}}$ is the activation output of the $(l-1)$-th layer. For $l = 1$, the initial activation is given by $a^{(0)} = x$.

Next, the network applies a predefined set of symbolic operators to the linear output $z^{(l)}$ to obtain the activation output $a^{(l)}$. These operators fall into two categories:

**Unary operators** $f_j : \mathbb{R} \to \mathbb{R}$, such as the identity function $\mathrm{id}(z) = z$, the sine function $\sin(z)$, and the exponential function $\exp(z)$. These operators depend on a single input variable and are used to model basic nonlinear or periodic behaviors. **Binary operators** $g_k : \mathbb{R}^2 \to \mathbb{R}$, such as the product operator $g(x, y) = x \cdot y$, which models interactions between pairs of inputs.

If the current hidden layer contains $u$ unary operators and $v$ binary operators, then the linear output dimension must satisfy $m_l = u + 2v$ in order to provide the required inputs.

The activation output of layer $l$ is then given by

$$a^{(l)} = \left[ f_1\left(z_1^{(l)}\right), \ldots, f_u\left(z_u^{(l)}\right), g_1\left(z_{u+1}^{(l)}, z_{u+2}^{(l)}\right), \right.$$
$$\left. \ldots, g_v\left(z_{m_l-1}^{(l)}, z_{m_l}^{(l)}\right) \right]^\top$$

where the first $u$ components are obtained by applying unary transformations, and the remaining $v$ components are produced by pairwise binary operations.

The output layer (layer $L$) does not apply any nonlinear symbolic operators and only performs a linear combination to produce the final prediction $\hat{y} \in \mathbb{R}$:

$$\hat{y} = E(x) = W^{(L)} a^{(L-1)} + b^{(L)},$$

where $W^{(L)} \in \mathbb{R}^{1 \times n_{L-1}}$ and $b^{(L)} \in \mathbb{R}$.

More generally, the nested computation of the entire network can be written as

$$E(x) = W^{(L)} \cdot \Lambda^{(L-1)}\Big(W^{(L-1)} \cdot \Lambda^{(L-2)}\big(\cdots$$
$$\Lambda^{(1)}\big(W^{(1)}x + b^{(1)}\big) \cdots + b^{(L-2)}\big) + b^{(L-1)}\Big) + b^{(L)}.$$

where each $\Lambda^{(l)}(\cdot)$ denotes the joint action of all unary and binary symbolic operators in layer $l$.

This nested expression highlights the structural characteristics of the EQL network: by composing multiple layers of linear mappings and symbolic functions, the model incrementally constructs an interpretable analytic expression. Compared with conventional neural networks, EQL not only approximates function values but also directly encodes the underlying functional structure, making explicit formula recovery possible.

### 3.2. Key Observations and Motivation

In a series of controlled experiments on target functions, we observe that even when the EQL network is initialized near an "almost optimal" structure that closely matches the target analytic expression, training quickly activates redundant operator paths (such as $\exp$) and eventually converges to a stable solution that contains redundant terms. When training is extended, this drift phenomenon often becomes more pronounced. These observations indicate that, under the existing architecture and training mechanism, the target analytic structure is not always a stable local minimum: redundant paths can continue to receive non-zero gradient signals, thereby undermining exact structure recovery. Figure 1 visualizes the network structure and parameter gradients at training steps 0, 1, and 1000, showing how redundant branches are "ignited" by gradients early on and may remain active instead of being naturally driven to zero near convergence. More details on the experimental setup, fitted expressions, and additional cases are provided in Appendix A.

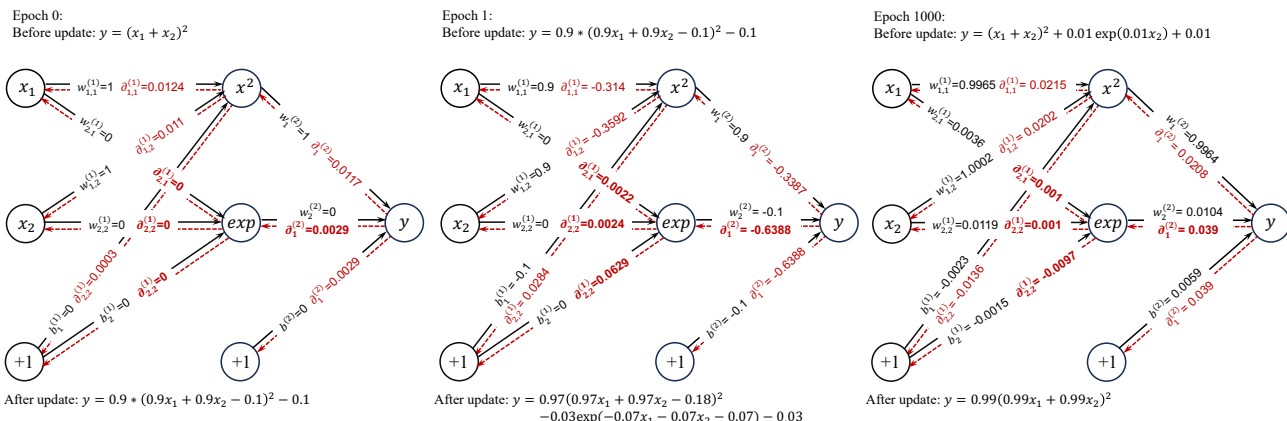

*Figure 1.* Network structure and corresponding gradients at epochs 0, 1, and 1000 during training.

## 4. Methodology

In this section, we present EQL-Z, a structurally controllable training framework for EQL networks. We view the redundant-activation failure of EQL as a problem that can be mitigated through two complementary routes. The first route is to reduce redundancy before optimization: at the structural level, we adopt a *small-to-large* structure search that starts from a compact seed network and gradually increases width and depth under explicit complexity control, thereby reducing the number of unnecessary operator paths exposed to training. The second route is to make unavoidable redundant paths easier to silence: at the operator level, we introduce zero-point constraints and zero-point consistency transformations, so that a redundant path naturally becomes silent when both its input and weight are zero. Once the structure is fixed, we further apply BFGS-based numerical fine-tuning to precisely optimize continuous parameters.

### 4.1. Zero-Point Constraint for EQL

#### 4.1.1. THEORETICAL ANALYSIS

In Section 3, we observed that even when an EQL network is initialized with a structure very close to the target expression, training still unintentionally activates additional symbolic operators and eventually converges to a local optimum containing redundant terms. Our gradient analysis further shows that certain symbolic operators (such as $\exp$) output a non-zero constant even when their input is zero (e.g., $\exp(0) = 1$), thus inducing non-zero gradients under non-zero upstream residuals even when their preceding coefficients are zero. This implies that, although the connection weights on a path are set to zero, the path can still be activated and updated during training, breaking the intended sparsity pattern of the network.

In practice, to control model complexity and promote compact representations, one typically imposes $\ell_1$ or $\ell_0$ regular-

ization to shrink the weights of redundant connections. As a connection weight $\theta_j$ approaches zero, the contribution of its output term to the overall loss diminishes, and the backpropagated gradients through this path also become small. Moreover, since the input $z_j$ to this path is usually a linear combination of the previous layer's outputs (e.g., $z_j = \sum_k w_{jk} a_k^{(l-1)}$), once the path is effectively shut down, the corresponding forward signal and backward gradient vanish. This, in turn, removes the update signal for the upstream weights $w_{jk}$, which are further driven toward zero by regularization. In the ideal case, this mechanism achieves path-level "structural silence."

However, if the operator itself produces a non-zero output at zero input, then even with zeroed weights, the path may still receive non-zero gradients and continue to be optimized. This mechanism provides one explanation for why redundant terms can be difficult to suppress in EQL-style models. Motivated by this, we propose to intervene at the level of operator definitions and introduce a *zero-point constraint*, requiring all symbolic operators $\Lambda(x)$ to satisfy $\Lambda(0) = 0$. Under the redundant-path condition where both the connection weight and the operator input are zero, this constraint makes the operator output zero and removes the residual gradient caused by the operator's non-zero response at the origin.

Taking the output layer as an example, if the weight of a certain connection is $\theta_j = 0$, the derivative of the model output with respect to this weight is given by

$$\frac{\partial \hat{y}}{\partial \theta_j} = \Lambda_j(z_j).$$

If at the same time $z_j = 0$ but $\Lambda_j(0) \neq 0$, then this path converts any non-zero upstream residual gradient into a non-zero update for the redundant connection. As a result, a connection that should be silent continues to be updated during training, undermining both sparsity and stability of

the learned expression. We formalize this phenomenon as follows:

**Definition 4.1** (Redundant operator path). In an EQL network, consider a connection represented as $\theta_j \cdot \Lambda_j(z_j)$, where $\theta_j$ is a trainable parameter, $\Lambda_j$ is a symbolic operator, and $z_j$ is its input. If, at the optimum, $\theta_j = 0$ and this path has no actual contribution to the output, we refer to it as a *redundant operator path*.

**Theorem 4.2** (Non-stationarity of redundant paths). *Let $\theta_j \cdot \Lambda_j(z_j)$ be a redundant path. If $\Lambda_j(0) \neq 0$, then the silent configuration $(\theta_j, z_j) = (0, 0)$ is generally not stationary during training, except when the upstream loss gradient vanishes.*

Conversely, when all operators satisfy the zero-point constraint $\Lambda(0) = 0$, the network acquires a natural structural compression property under the redundant-path condition: whenever a connection has zero weight and zero input, its path becomes silent with respect to this residual-gradient channel. This not only enhances sparsity but also improves the interpretability of the final expression. The above analysis shows that the zero-point constraint is not merely a minor design detail, but an important mechanism for improving training stability and model compression in this class of failure modes. The full mathematical derivation is provided in Appendix B.

### 4.1.2. ZERO-POINT CONSISTENT TRANSFORMATIONS

The zero-point constraint requires each symbolic function $\Lambda(x)$ to satisfy $\Lambda(0) = 0$, so as to avoid non-zero gradients on redundant connections when both the weight and input are zero (see Theorem 4.2). However, many common operators in standard symbolic libraries do not naturally satisfy this condition, e.g., $\exp(0) = 1$, $\cos(0) = 1$, and $\log(x)$ is not even defined at $x = 0$. Therefore, when constructing the operator set, we need to structurally transform such functions so that they exhibit consistent zero-point behavior.

We refer to this process as **zero-point consistent transformation**. Its goal is to enforce $\Lambda(0) = 0$ in a neighborhood of $x = 0$ without substantially altering the expressive power of the operator. Depending on the function, we consider two types of transformations:

**Output shift.** We construct $\Lambda(x) = f(x) - f(0)$, which is applicable to functions that are well-defined at $x = 0$, such as exp, cos, and power functions. **Input shift.** We construct $\Lambda(x) = f(x + \delta)$, which is suitable for cases where the original function is undefined or undesirable at $x = 0$, such as $\log(x)$.

These transformations do not systematically reduce the reachability of common target functions, but they can change the minimal parameterization of the recovered formula. For example, a target term $a \exp(bx) + c$ can be

represented with the transformed operator as $a(\exp(bx) - 1) + (a + c)$, where the constant term absorbs the output shift. Thus, zero-point transformations may introduce offset terms in the raw exported expression, which we simplify after symbolic extraction.

In practice, we recommend prioritizing output-shift transformations when designing symbolic network architectures, since they preserve the gradient propagation structure while enforcing zero response at the origin, allowing redundant connections to naturally decay during training. For operators that cannot be handled by output shifts (e.g., $\log(x)$), we instead apply input shifts that move the domain away from zero. For singular operators such as division, we follow the regularized division used in EQL÷ (Sahoo et al., 2018): for threshold $\theta > 0$,

$$h^\theta(a, b) = \begin{cases} a/b, & b > \theta, \\ 0, & \text{otherwise,} \end{cases} \qquad p^\theta(b) = \max(\theta - b, 0).$$

Here $h^\theta$ sets both the output and the corresponding gradients to zero in the forbidden small-denominator region, while $p^\theta$ penalizes such denominators and steers the network away from them. This treatment is consistent with our silent-path stability view, although a unified theory for all singular operators remains an important extension. Overall, these transformations act as a structural preprocessing step, decoupled from the training procedure, and systematically enhance sparsity and interpretability. Zero-point consistent transformations for common symbolic operators are summarized in Appendix B.3.

### 4.2. Small-to-Large Structure Search

While zero-point constraints make redundant paths easier to silence once they appear in a candidate structure, a complementary route is to reduce the number of redundant paths exposed to optimization in the first place. To this end, we propose a *small-to-large* structure search strategy: starting from a compact initial architecture $\mathcal{L}_0$, specified by the user or chosen heuristically, we gradually expand the network in width and depth through local modifications, while explicitly controlling search complexity and preserving expressive power. This structure-level route differs from training a large network and pruning afterward: it treats redundancy reduction as part of the search process itself, so new operators are introduced only when they improve a complexity-penalized validation score.

**Structure representation and stage control** Concretely, we represent an EQL architecture as a list of operator sets across layers,

$$\mathcal{L} = \{\mathcal{O}^{(1)}, \dots, \mathcal{O}^{(L)}\}, \qquad \mathcal{O}^{(l)} = \{o_1^{(l)}, \dots, o_{m_l}^{(l)}\},$$

and search within a predefined range of layer counts $[L_{\min}, L_{\max}]$ and per-layer operator caps $m_l \leq m_{\max}$. All operators are grouped into families $\{\mathcal{F}_1, \ldots, \mathcal{F}_S\}$ based on their functional roles or complexity (e.g., polynomial, trigonometric, logarithmic/rational, etc.). In our implementation, we introduce a stage variable $s$ to control progressive operator availability: at stage $s$, local growth samples operators from $\bigcup_{i \leq s} \mathcal{F}_i$, while higher-level operators and the associated candidate structures remain inactive until later stages are opened. The stage $s$ is increased according to heuristic rules based on validation performance gains and search progress (for example, progressively unlocking higher-level operator families when the performance improvement over a given number of steps becomes marginal), thereby preventing overly complex operators from being introduced too early.

**Local growth and complexity-penalized scoring**   At each step $t$, the algorithm constructs a compact candidate pool around the current structure $\mathcal{L}_t$, with *local growth* serving as the primary mechanism through two types of edits: (i) **Widening (AddOp).** For an existing layer $\ell$, we sample an operator from the families allowed at the current stage and add it to $\mathcal{O}_t^{(\ell)}$, provided that this layer has not yet reached the cap $m_{\max}$. (ii) **Deepening (AddLayer).** When $L_t < L_{\max}$, we append a new layer at the output side, typically initialized with a small number of newly sampled operators. In implementation, this candidate pool can be lightly augmented by bounded neighborhood edits, such as restricted operator replacement or small layer templates, while equivalent structures induced by operator permutations within the same layer are canonicalized and deduplicated. Thus, the search trajectory still moves through the structure space with very small step sizes, avoiding large jumps over a huge candidate set.

To achieve a controlled trade-off between accuracy and complexity, we define a complexity-penalized score for each structure $\mathcal{L}$:

$$S(\mathcal{L}) = R_{\text{val}}^2(\mathcal{L}) - \lambda_{\text{comp}} \cdot \frac{\text{Comp}(\mathcal{L})}{C_{\max}},$$

where $R_{\text{val}}^2$ denotes the coefficient of determination on the validation set, $\text{Comp}(\mathcal{L})$ is the symbolic-tree size or the number of operators, and $C_{\max}$ is set to the maximum allowed structural complexity under the current search budget, so that the complexity penalty is normalized across candidate architectures. For each evaluated candidate, we train the EQL model on the corresponding structure (using zero-point-consistent operators and a standard gradient-based optimizer) to obtain $R_{\text{val}}^2$ and the corresponding complexity-penalized score $S(\mathcal{L}_{\text{new}})$, which is then compared with the best historical score $S^\star$. We accept the structural update only if the score improves upon $S^\star$ by at least a small threshold $\delta_{\text{score}}$. For candidates that increase depth, we impose

an additional requirement that the validation gain $\Delta R^2$ exceed a preset threshold $\varepsilon_{\text{layer}}$, so as to prevent ineffective increases in depth.

**Family-adaptive sampling and early stopping**   The sampling over operator families is controlled by a set of learnable weights $\{w_i\}$. We treat the one-step performance improvement $\Delta R^2$ as a local reward for the currently selected family and update $w_i$ via an exponentially weighted moving average. We then apply a softmax with temperature scaling and lower-bound clipping to obtain family-level sampling probabilities $\{\pi_i\}$. In this way, operator families that have historically led to larger performance gains are more likely to be sampled in subsequent steps, gradually focusing the structure search on an "effective operator subspace."

The overall search terminates early when any of the following conditions is met: the maximum number of steps is reached, the score shows no significant improvement for $K_{\text{stop}}$ consecutive steps, or $R^2$ exceeds a preset threshold. The final output is the structure with the best trade-off between complexity and accuracy. The complete procedure is summarized in Appendix C.

### 4.3. BFGS-Based Numerical Fine-Tuning

Once the small-to-large structure search has determined the final symbolic architecture $\mathcal{L}_{\text{best}}$, the remaining uncertainty in the model primarily lies in the numerical precision of its continuous parameters. We therefore perform an additional, independent numerical fine-tuning step on these coefficients under a fixed structure, in order to further improve the fitting accuracy.

Concretely, we first extract from the EQL network the analytic expression corresponding to $\mathcal{L}_{\text{best}}$ and collect all continuous parameters appearing in the expression, including affine weights, biases, and scalar coefficients introduced by symbolic paths, into a parameter vector $\theta \in \mathbb{R}^d$. Given training data $(X, Y)$ (optionally augmented with a subset of validation data), we define the mean squared error loss

$$\mathcal{L}_{\text{MSE}}(\theta) = \frac{1}{N} \sum_{n=1}^{N} \left( f_{\mathcal{L}_{\text{best}}}(x_n; \theta) - y_n \right)^2,$$

where $f_{\mathcal{L}_{\text{best}}}$ denotes the symbolic network with fixed structure. We then minimize $\mathcal{L}_{\text{MSE}}$ in the continuous parameter space using BFGS, a second-order quasi-Newton method that employs local Hessian approximations to more accurately capture the second-order effect of parameter changes on the loss, thereby obtaining high-precision coefficient estimates in relatively few iterations.

In implementation, BFGS optimization is applied directly to the extracted analytic expression rather than re-entering the full neural network training loop: each iteration only

involves forward evaluation of the expression, backward computation of gradients with respect to $\theta$, and a parameter update according to the BFGS rule. This process is equivalent to a "last-mile" numerical refinement of the coefficients under a fixed symbolic form, bringing the model output closer to the true coefficients and constant terms of the target function. In our experiments, this step typically yields a noticeable improvement in $R^2$ without changing the structure and reduces the slight biases introduced by first-order optimizers.

# 5. Experiments

In this section, we systematically evaluate the effectiveness, stability, and structural compactness of the proposed method. Our experiments are organized around the following research questions (RQs), aiming to assess the performance of EQL-Z on function recovery tasks and to understand the contribution of its key components:

**RQ1.** On standard benchmark datasets, does EQL-Z achieve a better trade-off between predictive accuracy and model interpretability (e.g., expression complexity) than existing symbolic regression methods?

**RQ2.** Beyond the theoretical analysis, does the proposed zero-point constraint effectively suppress redundant operators in practice and improve training stability?

**RQ3.** To what extent does EQL-Z improve the exact equation recovery rate?

**RQ4.** How well does EQL-Z generalize and extrapolate under distribution shift and observational noise?

**RQ5.** How does EQL-Z perform on a real-world memory dataset? To answer RQ5, we include an auxiliary experiment on human memory modeling based on the MaiMemo dataset. The detailed setup and symbolic expressions are provided in Appendix G.

## 5.1. Experimental and Evaluation Setup

We evaluate EQL-Z on three types of datasets: synthetic Nguyen benchmark functions (Uy et al., 2011), SR-Bench (La Cava et al., 2021), and novel benchmark problems across three scientific domains (Shojaee et al., 2025). The Nguyen benchmarks assess exact equation recovery under controlled noise; SRBench includes both datasets with known analytic ground truth and purely black-box regression problems, which we use to assess the time-limited performance of EQL-Z under a tight computation budget; and the scientific benchmarks comprise four LLM-SR problems in nonlinear dynamics, microbiology, and materials science (Oscillation 1, Oscillation 2, *E. coli* growth, and Stress–Strain), jointly testing ID/OOD generalization on realistic physical systems. We consider three Gaussian noise

levels with standard deviations ($\sigma = 0.1, 0.01, and 0.001$). Following the range specifications in (Landajuela et al., 2022), we generate 12 Nguyen functions, each with 1000 training points. More details are provided in Appendix E. Unless otherwise specified, EQL-Z searches over the full operator library {id, square, cube, sin, cos_reg, mul, add, sqrt_reg, exp_reg, log_reg, pow_reg, div_reg}, including protected division rather than a customized smooth-only operator subset. As an additional stress test for division, EQL-Z achieves 100% exact recovery on the division-containing Feynman-I-18-4 formula, while Feynman-I-39-22 remains challenging due to its combined division and multiplicative numerator structure.

## 5.2. RQ1: Performance Comparison

Figure 2 compares EQL-Z with multiple baselines on the SRBench black-box datasets in terms of predictive accuracy, tolerance-based accuracy, and expression length across different noise levels. As shown, while EQL-Z is slightly behind some advanced GP-based methods, it maintains a much more compact structure with lower complexity. Compared with the original EQL, EQL-Z achieves consistently better overall performance and recovers substantially simpler models; its advantages are particularly evident in fine-grained metrics (e.g., per-instance errors), reducing the average model complexity by more than 80% on this benchmark. Moreover, EQL-Z is more robust to noise, outperforming the original EQL under all noise levels, with especially pronounced gains in low-noise settings. Ground-truth results are reported in Appendix F.

## 5.3. RQ2: Effectiveness of the Zero-Point Constraint

To verify whether the zero-point constraint suppresses unnecessary operator activation, we conduct a controlled comparison and track early-stage gradient responses of each operator (Fig. 3). We use the vanilla EQL architecture without our zero-point consistent transformation, pruning, or BFGS fine-tuning as a baseline. The target is $eq = (x_1 + x_2)^2$ with $x_1, x_2 \sim U(-1, 1)$ and Gaussian noise $\epsilon \sim \mathcal{N}(0, 0.01^2)$: target $= (x_1 + x_2)^2 + \epsilon$. We sample 100 points and train EQL with learning rate 0.1 for 20,000 steps using $L_1$ regularization (0.01). We compare training with and without the zero-point constraint ($\Lambda(0) = 0$) under identical initialization.

At epoch 0, gradients are almost identical; due to noise, the square operator shows a weak response in both settings. At epoch 1, *without* the constraint, the exp operator's gradient rapidly increases (peak $> 0.6$), indicating spurious activation. In contrast, *with* the constraint, the exp gradient remains zero throughout training, even in the early stage. This supports our claim that enforcing $\Lambda(0) = 0$ blocks invalid gradient paths and prevents redundant operators from

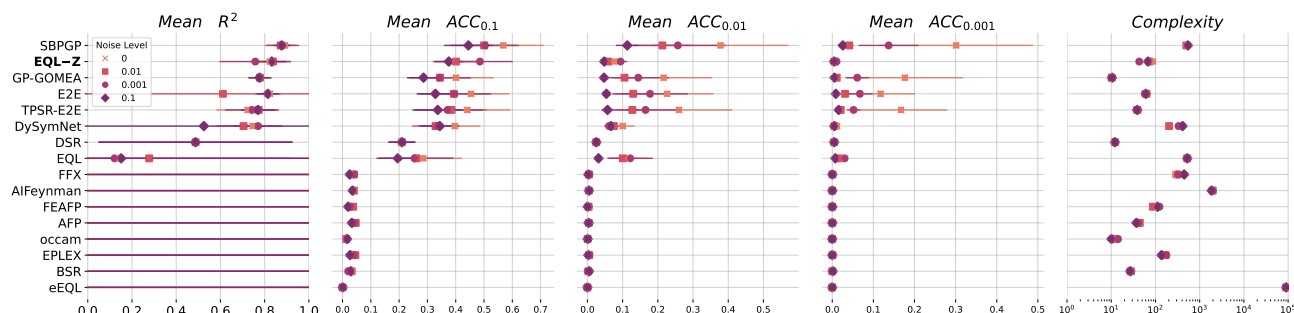

*Figure 2.* Performance and complexity comparison on black-box datasets, including $R^2$, tolerance-based accuracy, and expression length across noise levels. Each subplot shows algorithms sorted by their average $R^2$ scores across four noise levels, with points indicating averaged results and error bars indicating performance variance across subtasks.

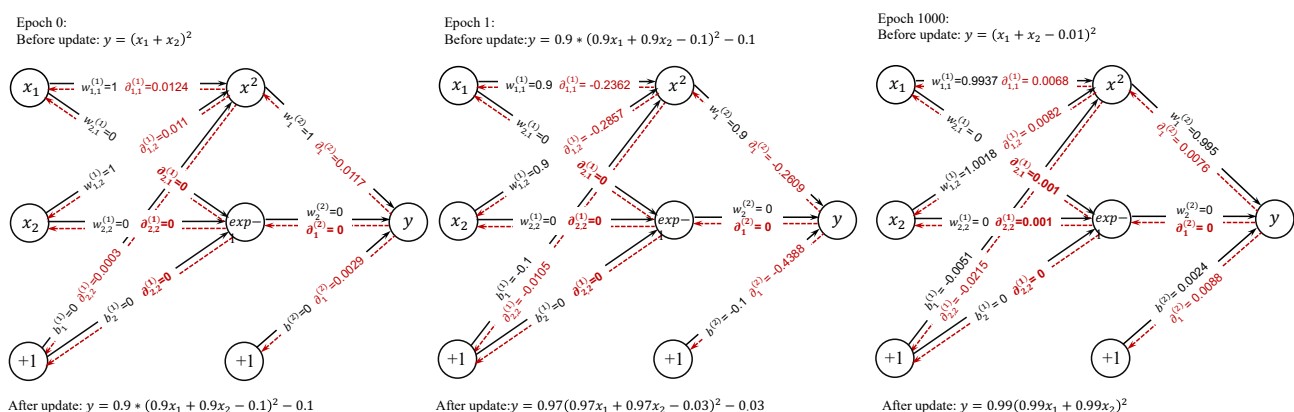

*Figure 3.* Network and gradient information for the target $(x_1 + x_2)^2$ after applying zero-point corrections to the operators.

*Table 1.* Exact equation recovery rates on the 12 standard Nguyen benchmarks. EQL-family results are averaged over 20 independent runs. Columns marked with † are taken from prior work (Scholl et al., 2025); full component-wise ablations are reported in Appendix F.

| Benchmark | Expression | EQL-Z | EQL | ParFam† | SPL† | NGGP† | GP† |
|-----------|-----------|-------|-----|---------|------|-------|-----|
| Nguyen-1 | $x^3 + x^2 + x$ | 100% | 0% | 100% | 100% | 100% | 99% |
| Nguyen-2 | $x^4 + x^3 + x^2 + x$ | 100% | 0% | 100% | 100% | 100% | 90% |
| Nguyen-3 | $x^5 + x^4 + x^3 + x^2 + x$ | 100% | 0% | 100% | 100% | 100% | 34% |
| Nguyen-4 | $x^6 + x^5 + x^4 + x^3 + x^2 + x$ | 100% | 0% | 100% | 99% | 100% | 54% |
| Nguyen-5 | $\sin(x^2)\cos(x) - 1$ | 80% | 0% | 83% | 95% | 80% | 12% |
| Nguyen-6 | $\sin(x) + \sin(x^2 + x)$ | 100% | 0% | 83% | 100% | 100% | 11% |
| Nguyen-7 | $\log(x+1) + \log(x^2+1)$ | 100% | 0% | 100% | 100% | 100% | 17% |
| Nguyen-8 | $\sqrt{x}$ | 100% | 0% | 100% | 100% | 100% | 100% |
| Nguyen-9 | $\sin(x_0) + \sin(x_1^2)$ | 100% | 0% | 100% | 100% | 100% | 76% |
| Nguyen-10 | $2\sin(x_0)\cos(x_1)$ | 100% | 0% | 100% | 100% | 100% | 86% |
| Nguyen-11 | $x^y$ | 100% | 0% | 0% | 100% | 100% | 13% |
| Nguyen-12 | $x_0^4 - x_0^3 + 0.5x_1^2 - x_1$ | 100% | 0% | 100% | 28% | 4% | 0% |
| Average | | 98.3% | 0.0% | 88.8% | 93.5% | 90.3% | 49.3% |

being mistakenly activated.

## 5.4. RQ3: Equation Recovery Experiments

We evaluate the exact equation recovery capability of different methods on the Nguyen benchmarks and perform an

ablation study on the zero-point constraint and the small-to-large structure search. For methods from the EQL family (vanilla EQL and our EQL-Z variants), the results are averaged over 20 independent runs with different random seeds, while the other baselines are taken directly from the ParFam paper (Scholl et al., 2025). Table 1 reports the equation

*Table 2.* Generalization performance on real-world tasks (ID/OOD NMSE; lower is better). Rows marked with † are taken from prior work (Shojaee et al., 2025).

| Model | Oscillation 1 | | Oscillation 2 | | *E. coli* growth | | Stress–strain | |
|---|---|---|---|---|---|---|---|---|
| | ID ↓ | OOD ↓ | ID ↓ | OOD ↓ | ID ↓ | OOD ↓ | ID ↓ | OOD ↓ |
| GPlearn† | 0.0155 | 0.5567 | 0.7551 | 3.1880 | 1.0810 | 1.0390 | 0.1063 | 0.4091 |
| NeSymReS† | 0.0047 | 0.5377 | 0.2488 | 0.6472 | N/A ($d \leq 3$) | N/A ($d \leq 3$) | 0.7928 | 0.6377 |
| E2E† | 0.0082 | 0.3722 | 0.1401 | 0.1911 | 0.6321 | 1.4467 | 0.2262 | 0.5867 |
| DSR† | 0.0087 | 0.2454 | 0.0580 | 0.1945 | 0.9451 | 2.4291 | 0.3326 | 1.1080 |
| uDSR† | 0.0003 | 0.0007 | 0.0032 | 0.0015 | 0.3322 | 5.4584 | 0.0502 | 0.1761 |
| PySR† | 0.0009 | 0.3106 | 0.0002 | 0.0098 | 0.0376 | 1.0141 | 0.0331 | 0.1304 |
| EQL | 2.5078 | 2.3894 | 0.6686 | 1.6729 | 0.9709 | 1.0022 | 0.3377 | 1.0349 |
| **EQL-Z** | 0.0032 | 0.0087 | 0.0057 | 0.0036 | 0.8405 | 0.9253 | 0.0346 | 0.1597 |
| LLM-SR (Mixtral)† | **7.89e-8** | **0.0002** | 0.0030 | 0.0291 | **0.0026** | **0.0037** | **0.0162** | 0.0946 |
| LLM-SR (GPT-3.5)† | **4.65e-7** | **0.0005** | **2.12e-7** | **3.81e-5** | **0.0214** | **0.0264** | **0.0210** | **0.0516** |

recovery rates on the 12 standard Nguyen functions, and the complete component-wise 20-run ablation is provided in Appendix F (Table 6). We observe that EQL-Z achieves 100% recovery on 11 out of the 12 benchmarks and reaches 80% on the most challenging case, Nguyen-5, yielding an average recovery rate of 98.3%, which is substantially higher than the 0.0% of vanilla EQL under the same protocol.

The ablation results in Appendix F show that removing the zero-point constraint (EQL-Z$_{\text{w/o ZP}}$) reduces the average recovery to 86.3%, while further discarding the small-to-large strategy in favor of a single large, fixed architecture (EQL-Z$_{\text{w/o STL}}$) drives the recovery rate to 0% on several functions and reduces the average recovery to 17.9%. Overall, EQL-Z achieves recovery performance comparable to strong baselines such as ParFam (88.8%) and SPL (93.5%), and together with Appendix Table 5 and Table 6, these results highlight that zero-point-consistent operators, small-to-large structure search, and numerical refinement play complementary roles in equation recovery. More specifically, the small-to-large strategy provides the main structural benefit by reducing the chance that interfering operators enter the candidate architecture, while the zero-point constraint improves optimization robustness when redundant operators are still present in a structure. BFGS serves a different role: it mainly refines numerical coefficients after a promising symbolic form has already been found, rather than performing structural discovery. This component-wise behavior supports our view that EQL-Z improves recovery through both structure-level search control and operator-level gradient stabilization.

### 5.5. RQ4: Generalization Experiments

Table 2 shows that LLM-SR achieves the lowest overall ID and OOD errors across the four real-world tasks. Among methods without language priors, EQL-Z remains competitive while substantially improving over the original EQL, often by one or more orders of magnitude in ID/OOD NMSE.

## 6. Conclusion

We introduced EQL-Z, a structurally controllable neuro-symbolic regression framework that improves the stability of EQL training by addressing a residual-gradient mechanism with zero-point constrained operators, small-to-large structure search, and BFGS fine-tuning on extracted analytic expressions. Experiments across synthetic, black-box, physical, and real-world behavioral tasks show that EQL-Z improves equation recovery and expression compactness over the original EQL, while remaining competitive with leading symbolic regression baselines. Future work includes extending the operator library and combining EQL-Z with automated architecture search to handle richer operator sets and more complex symbolic spaces.

## Acknowledgements

This work was financially supported by the National Natural Science Foundation of China (62437002, 62577028, 62293554), Hubei Provincial Natural Science Foundation of China (2023AFA020), and Fundamental Research Funds for the Central Universities (JC2026TS-014, XJ2026001102, XJ2026003301).

## Impact Statement

This work aims to advance the field of symbolic regression and interpretable modeling by improving the stability and controllability of EQL-like methods. We expect its societal impact to be similar in nature to that of existing machine learning tools for scientific discovery and data analysis, and we are not aware of any immediate, domain-specific risks beyond the usual concerns about data quality, bias, and misuse.

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

# A. Experimental Case

## A.1. Network Illustrations

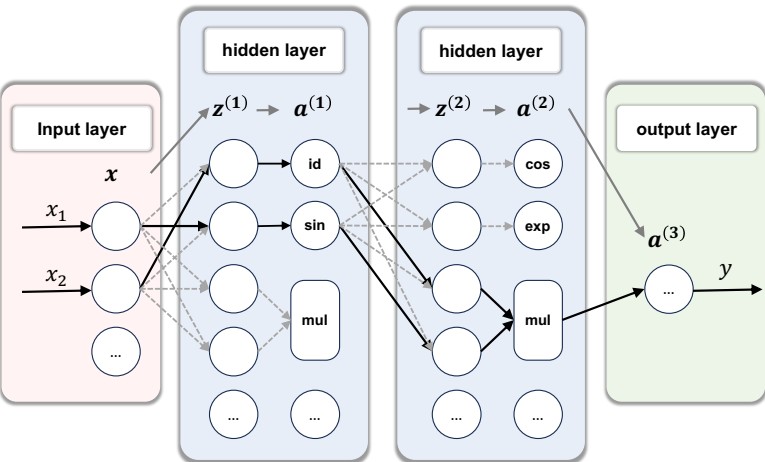

*Figure 4.* EQL network. The figure illustrates a 3-layer EQL network, where each hidden layer contains three operators: two unary operators and one binary operator ($u = 2, v = 1$). Solid lines indicate the target equation $y = x_2 \sin(x_1)$.

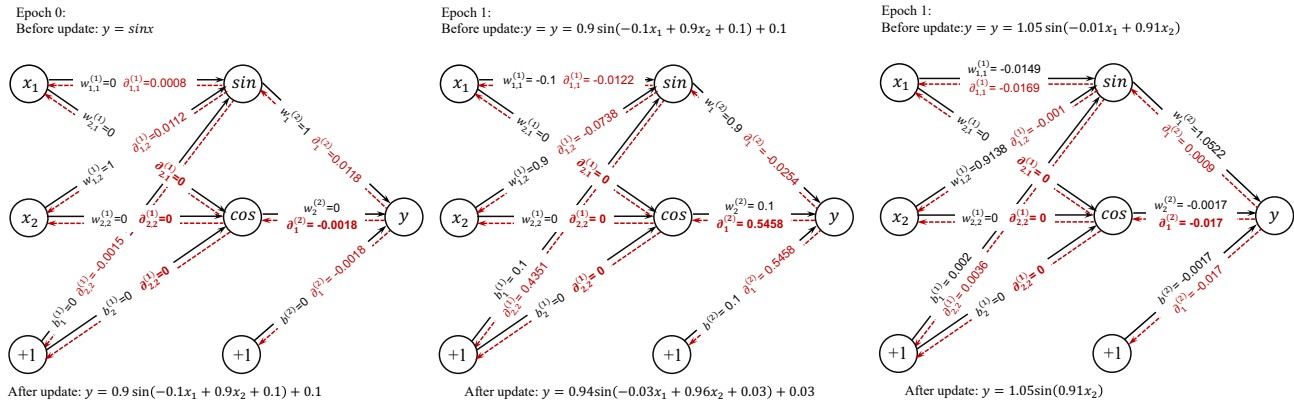

*Figure 5.* Network structure and corresponding gradients during training for the target function $\sin(x_2)$.

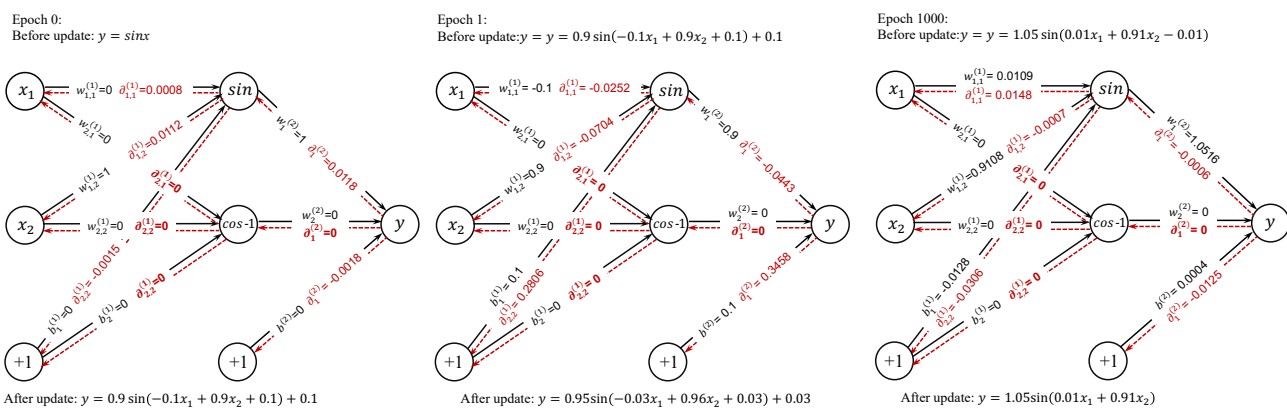

*Figure 6.* Network structure and corresponding gradients during training for $\sin(x_2)$ after applying operator corrections.

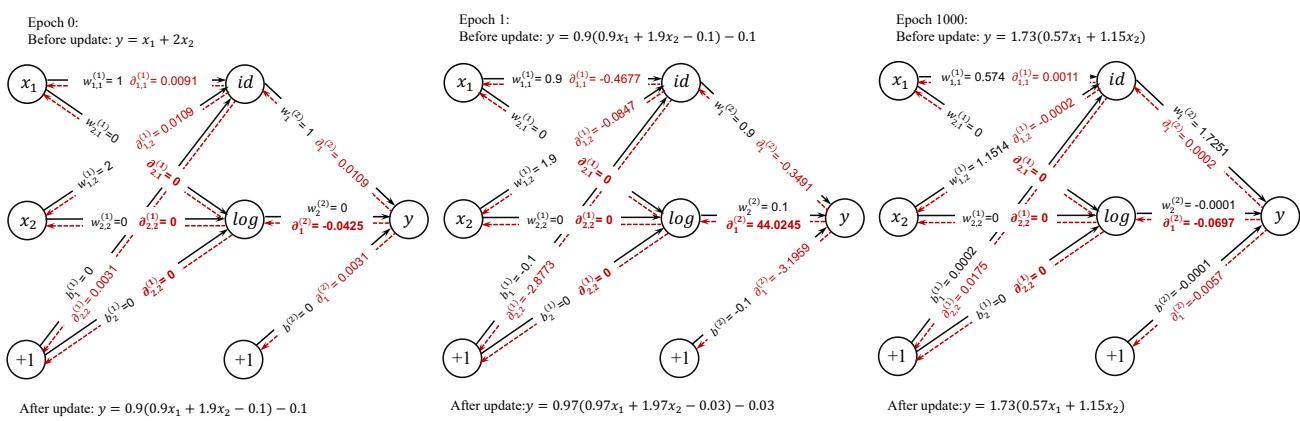

*Figure 7.* Network structure and corresponding gradients during training for the target function $x_1 + 2x_2$.

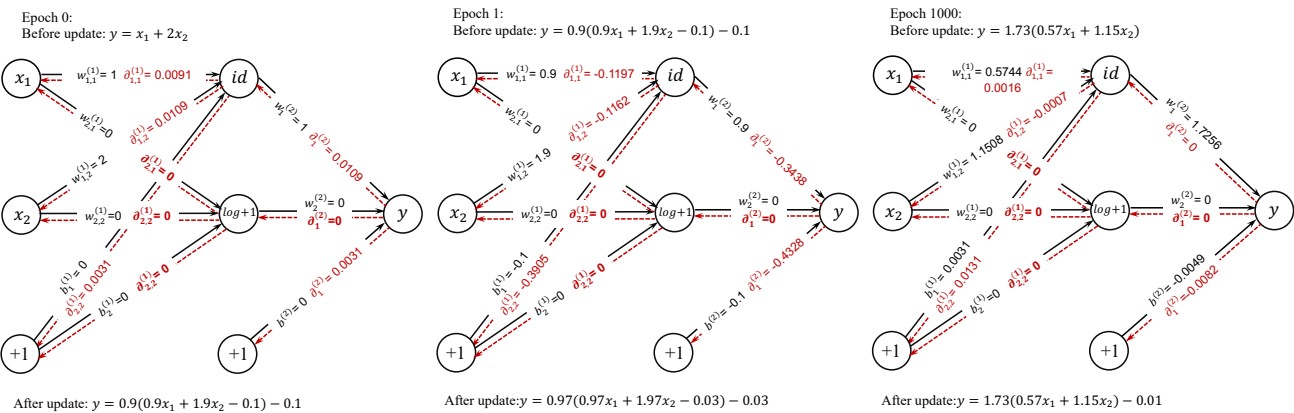

*Figure 8.* Network structure and corresponding gradients during training for $x_1 + 2x_2$ after applying operator corrections.

## A.2. Experimental Setup

We first consider a simple toy problem where the ground-truth target formula is eq $= (x_1 + x_2)^2$. We sample $x_1$ and $x_2$ independently from the uniform distribution $U(-1, 1)$ and generate 100 data points in total. Gaussian noise with standard deviation 0.01 is added to the target values:

$$x_1, x_2 \sim U(-1, 1), \quad \epsilon \sim \mathcal{N}(0, 0.01^2),$$
$$\text{target} = (x_1 + x_2)^2 + \epsilon.$$

The goal of the EQL network is to represent this formula as a neural architecture and progressively approximate it through layer-wise compositions of symbolic operators. During training, the network adjusts its parameters to recover the underlying analytic structure from noisy observations.

With an $\ell_1$ regularization coefficient $\lambda_{L_1} = 0.01$, we use an EQL architecture with a single hidden layer whose operator set is $[[\texttt{square}, \exp]]$. Despite the simplicity of the target expression and the fact that the network has sufficient representational capacity, the model still fails to recover the desired structure. For this operator configuration, the final fitted expression is

$$\hat{y} = 0.192 + 0.094 \cdot \exp(-1.882x_1) \exp(-2.042x_2).$$

Clearly, the model does not reconstruct the quadratic structure $(x_1 + x_2)^2$; instead, it concentrates on an exponential-path representation. This indicates that, even when the architecture and hyperparameters are well aligned with the target, EQL can converge to an incorrect solution and struggle to reconstruct the intended analytic form. We hypothesize that this failure stems from the optimization trajectory not following a path toward the simplest compatible structure.

To probe this issue, we design an extreme yet instructive experiment: we initialize the network to be almost identical to the target formula,

$$1.0 \cdot (1.0x_1 + 1.0x_2)^2,$$

with all bias terms and exponential-related parameters set to zero in order to eliminate interference from unnecessary operators. Although such initialization is nearly impossible to obtain in practical applications, it should in principle provide an ideal starting point for optimization.

From an empirical standpoint, when a model is initialized very close to the target analytic structure, one would expect it to quickly converge to the ideal solution. In our experiment, the model triggers early stopping within 1000 epochs, and the final fitted expression (rounded to three decimal places) is

$$\hat{y} = 0.979 \cdot (0.995x_1 + x_2 + 0.004)^2$$
$$+ 0.006 \cdot \exp(-0.004x_1 + 0.014x_2).$$

The activation of the exponential term indicates that the network still introduces redundant structure. More importantly, when we disable early stopping and continue training for longer, the deviation from the target structure becomes even more pronounced. This suggests that, even when starting from a near-optimal configuration, EQL can be easily guided toward non-analytic paths during training, revealing an inherent lack of structural stability.

### A.3. Gradient Analysis

To analyze the gradient behavior at both the early and late stages of optimization, we record the parameter gradients at epochs 0, 1, and 1000 (see Figure 1).

Starting from an initialization that is already close to the theoretical optimum, we observe the following key phenomena in the evolution of gradients:

1. **Redundant terms are rapidly activated in the early stages.** Although the coefficients of the $\exp$ terms are initialized to zero, their gradients become non-zero after just one update step, indicating that the network immediately responds to these redundant operators. After a few steps, these gradients are further amplified (e.g., reaching a magnitude of about $-0.67$ in `layer_0`), showing that the model quickly drifts away from the compact structure and moves toward more complex expressions.

2. **The network converges to a local optimum containing redundant terms.** By epoch 1000, all gradients have dropped below the order of $10^{-4}$, indicating that the model is close to convergence. However, the output still contains a non-negligible exponential term, which implies that the network has not converged to the optimal structure corresponding to the target analytic expression, but instead stabilizes at a local minimum with redundant components. In other words, even when starting near the theoretical solution, the current EQL network is still prone to being trapped in an incorrect, over-parameterized optimum.

We further conduct controlled experiments on two additional target formulas, $\sin(x_2)$ and $x_1 + 2x_2$ (see Appendix A), and obtain qualitatively similar results: even when the initialization is very close to the target expression, redundant operators are activated during training and are not naturally suppressed to zero. These observations point to a core issue: under the current architecture and training mechanism, the ground-truth analytic formula does not correspond to a stable local minimum. Even when starting in the vicinity of the theoretical optimum, redundant operators become activated and maintain significant gradients over time, ultimately causing the network to converge to local optima that include superfluous terms.

## B. Proofs of Theorem 4.2

In this section, we provide a formal proof of Theorem 4.2, showing that if the symbolic operator $\Lambda_j$ on a redundant path does not satisfy the zero-point consistency condition (i.e., $\Lambda_j(0) \neq 0$), then the silent configuration $(\theta_j, z_j) = (0, 0)$ is generally not stationary during training unless the upstream loss gradient vanishes.

In practice, to control model complexity and encourage compact expressions, one typically introduces $\ell_1$ or $\ell_0$ sparsity regularization to drive the weights $\theta_j$ of redundant connections toward zero. Once a path has been compressed (i.e., $\theta_j \to 0$), its output term $\theta_j \cdot \Lambda_j(z_j)$ contributes negligibly to the overall error. Furthermore, the effect of regularization propagates

backward: since $z_j$ is usually a weighted sum of the previous layer's activations, e.g., $z_j = \sum_k w_{jk} a_k^{(l-1)}$, and the path is no longer used, the gradient contributions of this term to the loss also vanish. As a result, the upstream weights $w_{jk}$ connected to this path are also driven toward zero by sparsity regularization.

In other words, once a connection becomes redundant and its output is effectively pruned away, the gradients along the corresponding backpropagation chain tend to zero, causing the upstream weights to lose update signals. This in turn leads to $z_j \to 0$, yielding path-level *structural silence*.

However, this silencing mechanism critically depends on the behavior of $\Lambda_j$ at zero. If $\Lambda_j(0) \neq 0$, then even when $\theta_j = 0$ and $z_j = 0$, the gradient

$$\frac{\partial L_{\text{total}}}{\partial \theta_j} = \Lambda_j(0) \cdot \frac{\partial L_{\text{total}}}{\partial \hat{y}}$$

remains non-zero whenever the upstream loss gradient is non-zero, which breaks the intended decay of the path and induces unnecessary parameter perturbations and expression errors.

Therefore, we introduce a **zero-point consistency constraint** at the operator level, requiring all symbolic operators to satisfy

$$\Lambda(0) = 0.$$

Under the condition that a redundant connection has both zero weight and zero input, zero-point consistency makes the operator output zero and removes this residual-gradient channel along that path. As a consequence, redundant connections can more easily remain silent and be pruned, thereby enhancing network sparsity and controllability of the learned expressions.

### B.1. Non-zero Gradient Analysis

Consider the contribution of a single path in the network,

$$y_j = \theta_j \cdot \Lambda_j(z_j),$$

where $\theta_j$ is a trainable parameter and $\Lambda_j(z_j)$ is the output of a symbolic operator applied to its input.

At the point $(\theta_j, z_j) = (0, 0)$, we have

$$y_j = 0.$$

Assume the total loss $L_{\text{total}}$ depends on the model output $\hat{y} = \sum_j y_j$ and is differentiable with respect to $\theta_j$. Then at this point, the partial derivative of the loss with respect to $\theta_j$ is

$$\frac{\partial L_{\text{total}}}{\partial \theta_j} = \frac{\partial L_{\text{total}}}{\partial \hat{y}} \cdot \frac{\partial \hat{y}}{\partial \theta_j} = \frac{\partial L_{\text{total}}}{\partial \hat{y}} \cdot \Lambda_j(0).$$

Since $\frac{\partial L_{\text{total}}}{\partial \hat{y}} \neq 0$ in general during training (unless the output already perfectly fits the target), whenever $\Lambda_j(0) \neq 0$ we obtain

$$\frac{\partial L_{\text{total}}}{\partial \theta_j} \neq 0.$$

Hence, under a non-zero upstream loss gradient, the point $(\theta_j, z_j) = (0, 0)$ does not satisfy the zero-gradient condition for a stationary point, which proves the theorem. $\qquad\square$

### B.2. Necessity of Zero-Point Consistency in Pruning

This result highlights a key aspect of pruning: if the symbolic operator on a pruned path produces a non-zero output at zero input, then even after the path weight is set to zero, it will still induce a non-zero gradient during optimization and thus cannot remain naturally silent during training. To ensure that the optimization landscape remains stable after pruning, paths to be removed must satisfy

$$\Lambda_j(0) = 0.$$

Therefore, the silencing of redundant paths depends not only on driving the weights to zero, but also critically on the zero-point behavior of the operators themselves.

*Table 3.* Zero-point-consistent transformations for symbolic operators.

| Raw operator | $\Lambda_{\text{raw}}(0)$ | Transformed $\Lambda(x)$ | $\Lambda(0)$ | Type of treatment |
|---|---|---|---|---|
| $\text{id}(x)$ | 0 | $x$ | 0 | Satisfied natively |
| $\text{square}(x)$ | 0 | $x^2$ | 0 | Satisfied natively |
| $\text{cube}(x)$ | 0 | $x^3$ | 0 | Satisfied natively |
| $\sin(x)$ | 0 | $\sin(x)$ | 0 | Satisfied natively |
| $\cos(x)$ | 1 | $\cos(x) - 1$ | 0 | Output shift |
| $\exp(x)$ | 1 | $\exp(x) - 1$ | 0 | Output shift |
| $\log(x)$ | $-\infty$ | $\log(x + 1)$ | 0 | Input shift |
| $\text{mul}(x, y)$ | 0 | $x \cdot y$ | 0 | Satisfied natively |
| $\text{add}(x, y)$ | 0 | $x + y$ | 0 | Satisfied natively |
| $\text{pow}(x, y)$ | 1 | $\text{pow}(x, y) - 1$ | 0 | Output shift |
| $\text{div}(x, y)$ | undefined | $\begin{cases} x/y, & y > \theta \\ 0, & \text{otherwise} \end{cases}$ | 0 | Regularized division |

## B.3. Additional Notes: Operator Design Guidelines

Table 3 summarizes the zero-point-consistent transformations applied to common symbolic operators in EQL-Z.

## C. Small-to-Large Structure Search

Algorithm 1 outlines the small-to-large structure search procedure used in EQL-Z to grow compact architectures.

## D. Limitations and Future Work

The small-to-large structure search introduces additional computational cost and depends on heuristic choices such as seed structures, growth rules, and complexity-penalized scoring. Its scalability to larger input dimensions or substantially more complex operator libraries therefore remains an important direction for future work. In addition, although EQL-Z substantially improves over vanilla EQL on real-world scientific tasks, it still lags behind LLM-SR, suggesting that combining EQL-style continuous optimization with stronger structure priors is a promising direction.

## E. Experimental and Evaluation Setup

### E.1. Metrics and Baselines

We organize our evaluation metrics around the main research questions (RQs) in the paper, following established experimental protocols in symbolic regression and extrapolation evaluation.

**Metrics for RQ1: Predictive Performance and Model Compactness**    For **RQ1** (overall predictive performance and model compactness), we follow the experimental guidelines in (Biggio et al., 2021; Kamienny et al., 2022) and adopt three metrics: the coefficient of determination $R^2$, the tolerance-based accuracy $ACC_\omega$, and the expression complexity $Complexity$.

The $R^2$ score measures the goodness-of-fit between predictions $\tilde{y}_i$ and ground-truth values $y_i$:

$$R^2 = 1 - \frac{\sum_{i=1}^{n}(y_i - \tilde{y}_i)^2}{\sum_{i=1}^{n}(y_i - \bar{y})^2},$$

where $\bar{y}$ is the mean of the targets $y_i$.

The tolerance-based accuracy $ACC_\omega$ is defined as the fraction of samples whose relative error does not exceed a threshold $\omega$:

$$\frac{|y_i - \tilde{y}_i|}{|y_i|} \leq \omega,$$

---

**Algorithm 1** Small-to-Large Structure Search

---

**Input:** training/validation data $(X, Y)$; operator families $\{\mathcal{F}_1, \ldots, \mathcal{F}_S\}$; initial structure $\mathcal{L}_0$; hyperparameters $\Theta$ (depth/width limits, step budget, stage settings, early-stopping thresholds, etc.)

**Output:** selected structure $\mathcal{L}_{\text{best}}$ and its corresponding analytic expression

**Initialization:**

Set current structure $\mathcal{L} \leftarrow \mathcal{L}_0$

Train EQL on $\mathcal{L}$ (with zero-point-consistent operators) to obtain $R^2_{\text{curr}}$ and score $S(\mathcal{L})$

Set $S^\star \leftarrow S(\mathcal{L})$ and $\mathcal{L}_{\text{best}} \leftarrow \mathcal{L}$

Initialize family weights $w$, stage $s \leftarrow 1$, counters, and $t \leftarrow 0$

**repeat**

  $t \leftarrow t + 1$

  **(1) Sample family and operator:**

  Compute sampling probabilities $\pi$ for the current stage from $w$

  Sample a family index $i$ according to $\pi$, then sample an operator $o^\star \in \mathcal{F}_i$

  Use local growth (ADDOP/ADDLAYER) to build candidate structures around $\mathcal{L}$

  Optionally augment this pool with bounded neighborhood candidates, then select one candidate structure $\mathcal{L}_{\text{new}}$ for evaluation

  **(2) Train and score:**

  Train EQL on $\mathcal{L}_{\text{new}}$ to obtain $R^2_{\text{new}}$ and score $S_{\text{new}}$

  Compute $\Delta R^2 \leftarrow R^2_{\text{new}} - R^2_{\text{curr}}$

  **(3) Acceptance and state update:**

  **if** deepening **and** $\Delta R^2 < \varepsilon_{\text{layer}}$ **then**

    Reject this growth (keep $\mathcal{L}$)

  **else**

    **if** $S_{\text{new}} \geq S^\star + \delta_{\text{score}}$ **then**

      Accept candidate: update $\mathcal{L} \leftarrow \mathcal{L}_{\text{new}}$, $R^2_{\text{curr}} \leftarrow R^2_{\text{new}}$

      Update $S^\star \leftarrow S_{\text{new}}$ and $\mathcal{L}_{\text{best}} \leftarrow \mathcal{L}$

    **else**

      Reject candidate (keep $\mathcal{L}$)

    **end if**

  **end if**

  Update family weight $w_i$ using $\Delta R^2$

  Update stage/global stagnation counters; if stage-unlock holds then set $s \leftarrow s + 1$

**until** $t \geq T_{\text{max}}$ **or** early stopping is triggered **or** $R^2_{\text{curr}} \geq R^2_{\text{max}}$

**Return:** $\mathcal{L}_{\text{best}}$ and its corresponding analytic expression

---

i.e., $ACC_\omega$ is the proportion of samples that fall within this tolerance band (in the main text we report, e.g., $ACC_{0.1}$ and $ACC_{0.01}$).

The expression complexity $Complexity$ is measured by the formula length, defined as the total number of symbols and parameters in the recovered expression, following (Biggio et al., 2021; Kamienny et al., 2022).

**Metrics for RQ3: Exact Equation Recovery Rate** For **RQ3** (exact equation recovery ability), we adopt the recovery-rate protocol from recent controllable SR work (Scholl et al., 2025). The *equation recovery rate* is defined as the fraction of runs in which the recovered expression is algebraically equivalent to the ground-truth target.

Concretely, we consider a run successful if the recovered expression is algebraically equivalent to the target after simplification (e.g., by expansion, factorization, or logarithmic identities), even if their surface syntactic forms differ. For example,

$$\log(x^3 + x^2 + x^2 + x + 1) \quad \text{and} \quad \log(x^2 + 1) + \log(x + 1)$$

are treated as equivalent after algebraic simplification, and both count as "correct recovery". All reported recovery rates are averaged over multiple random seeds, initial structures, and initialization schemes.

*Table 4.* SR methods evaluated in this study and their classification. *Gen* denotes generative SR; *RL*, *GS*, and *GP* denote reinforcement learning, grid search, and genetic programming under exploratory SR, respectively.

| METHOD | CLASS | DESCRIPTION |
|---|---|---|
| TPSR-E2E | GEN | IMPROVED MCTS-STYLE OPTIMIZER FOR PRE-TRAINED SR MODELS |
| E2E | GEN | PRE-TRAINED MODEL FOR DIRECT FORMULA PREDICTION |
| DSR | RL | RNN-BASED REINFORCEMENT-LEARNING SR |
| AIFEYNMAN | GS | ENHANCED TOOLKIT FOR DISCOVERING PHYSICAL EQUATIONS |
| BSR | GS | BAYESIAN SYMBOLIC REGRESSION |
| FFX | GS | FAST FUNCTION EXTRACTION |
| EQL | GS | MLP WITH SYMBOLIC OPERATORS AS ACTIVATIONS |
| OCCAMNET | GS | SPARSE NEURAL SYMBOLIC MODEL WITH STOCHASTIC OPERATOR-PATH SELECTION |
| DYSYMNET | GS | EQL-STYLE MODEL WITH RL-DRIVEN STRUCTURE OPTIMIZATION |
| eEQL | GS | EVOLVING EQUATION LEARNER WITH PROGRESSIVE STRUCTURE EVOLUTION |
| AFP | GP | AGE–FITNESS PARETO OPTIMIZATION |
| AFP_FE | GP | AFP WITH CO-EVOLVED FITNESS ESTIMATES |
| EPLEX | GP | $\epsilon$-LEXICASE SELECTION |
| GP-GOMEA | GP | GP WITH THE GENE-POOL OPTIMAL MIXING EVOLUTIONARY ALGORITHM |
| SBP-GP | GP | SEMANTIC BACK-PROPAGATION GP |

**Metrics for RQ4: Generalization Under Distribution Shift**  For **RQ4** (generalization under distribution shift and extrapolation), we follow the in-/out-of-distribution (ID/OOD) evaluation protocol used in LLM-based symbolic regression (Shojaee et al., 2025) and use the *normalized mean squared error* (NMSE) as our main metric.

Given predictions $\tilde{y}_i$ and ground-truth values $y_i$, NMSE is defined as

$$\text{NMSE} = \frac{\sum_{i=1}^{n}(y_i - \tilde{y}_i)^2}{\sum_{i=1}^{n}(y_i - \bar{y})^2},$$

where the denominator normalizes the MSE by the target variance, making errors comparable across datasets.

**Baseline Methods**  To ensure a comprehensive comparison, we select representative or state-of-the-art methods from each family (GP-based, RL-based, EQL-style, and generative SR), yielding 15 baselines in total. These methods and their categorization are summarized in Table 4.

### E.2. Additional Experimental Details

**(1) Training Configuration and Compute Budget**  For all EQL-family methods, we use a unified training configuration rather than tuning separate hyperparameters for each Nguyen function. Unless otherwise specified, the learning rate is set to 0.01, the $L_1$ regularization coefficient is 0.01, and each candidate model is trained for 10,000 epochs. During the small-to-large structure search, we set $L_{\text{start}} = 1$, $L_{\text{max}} = 2$, max_ops_per_layer=3, operator_level_start=0, and operator_level_max=5. The search is initialized from several compact seed structures, including [[id]], [[id, square]], [[id, square, cube]], and [[sin]].

The main experiments are run on an RTX 4090 with 15 parallel jobs and a 1200-second time budget for each task. In practice, EQL-Z evaluates about 50 candidate structures on average during the small-to-large search, and the wall-clock runtime ranges from roughly 10 minutes to 6 hours depending on task difficulty and search progress. For the SRBench black-box setting, we use a constrained 10-minute budget per dataset, so these results should be interpreted as short-budget performance rather than a reproduction of the long-horizon SRBench protocol.

**(2) Three Initialization Schemes**  In all comparisons, we use three initialization schemes for EQL-type models: default, ones, and scaled.

**default initialization** samples all linear weights from a zero-mean Gaussian distribution with small variance and sets all biases to 0, corresponding to standard MLP initialization. This serves as a natural baseline but does not explicitly account

for the scale of symbolic operator outputs or inter-layer activation distributions.

**ones initialization** sets all weights to 1 and all biases to 0, so all connections contribute equally at the beginning of training. We mainly use this scheme to study the sensitivity of operator selection, pruning, and zero-point constraints under "identical initial coefficients and fixed structure", rather than as a performance-optimized initialization.

**scaled initialization (layer-wise RMS-calibrated initialization)** aims to keep the RMS scale of pre-activations and operator outputs within a controlled range across layers and operator blocks, without changing the network topology or operator set. This improves training stability and reduces the risk that some operators are "over-ignited" at the very beginning.

Concretely, scaled initialization consists of two steps:

*(i) Per-layer scaling.* For each linear layer, we first construct a set of "probe samples" (e.g., by sampling from the data domain $[x_{\min}, x_{\max}]$ or uniformly from $[-1, 1]$) and pass them through the linear mapping to obtain pre-activations $z$. We then group outputs into "operator blocks" (e.g., columns associated with the same unary or binary operator family), compute the RMS of each block, and rescale the corresponding rows of the weight matrix so that each block's RMS matches a target scale (e.g., around 1 or a family-specific factor). This ensures each operator block has a reasonable numeric scale before its nonlinear symbolic operator is applied.

*(ii) Recursive cross-layer scaling.* After calibrating one layer, we apply its symbolic operators to the scaled pre-activations to obtain new activations, and then treat these activations as inputs to the next layer, repeating the same procedure. This top-down recursive calibration constrains the scales of linear mappings and symbolic outputs across depth, mitigating issues such as "exploding activations in a middle layer" or "near-zero activations in deeper layers".

Overall, scaled initialization provides a "structure-aware" pre-normalization of activation distributions without changing architecture or operator sets, stabilizing gradient propagation and, together with zero-point constraints, further suppressing spurious activation of redundant operators. Unless otherwise noted, all EQL(-Z) results in the main text are averaged over multiple runs under these three initialization schemes.

**(3) Multiple Initial Structures and Repeated Runs**   In the equation recovery experiments, we design multiple starting operator structures for EQL-Z to reduce bias from a single initial architecture. Typical starting structures include:

- Simple single-layer structures, such as $[[\text{id}]]$ and $[[\sin]]$;

- Lightweight polynomial structures, such as $[[\text{id}, \text{square}]]$;

- Several mixed structures combining basic polynomial and trigonometric operators.

For each candidate starting structure and each initialization scheme (default, ones, scaled), we run 10 independent trials with different random seeds. All reported recovery rates and performance metrics are computed from these repeated runs.

**(4) Role of Structure Choice and the Small-to-Large Strategy**   The core role of the small-to-large structure search strategy is not only to reduce the search space, but also to help the model automatically discover operator-layer structures that match the target task, thereby recovering more compact expressions that are closer to the ground truth.

Empirically, we find that directly using a large, fixed architecture with many operators (e.g., wide layers stacked in one shot) often leads EQL-like models to activate numerous redundant terms, even when the operator library is expressive enough to represent the target. The resulting expressions tend to be overly complex and less interpretable. In contrast, when we start from a compact architecture and progressively grow it by local widening and deepening, EQL-Z more reliably converges to reasonable operator combinations and layer layouts during the search.

For example, on some Nguyen functions, the structure search, after several local growth steps, converges to a two-layer architecture of the form

$$[[\text{id}, \text{square}, \text{cube}], [\text{log\_reg}]],$$

where the first layer consists of polynomial basis operators and the second layer contains a single logarithmic operator. Under this structure, EQL-Z is able to fully recover the target expression.

*Table 5.* Ablation study on the Nguyen benchmarks.

| Method | R2 | ACC01 | ACC001 | Complexity |
|---|---|---|---|---|
| EQL | $0.998 \pm 0.000$ | $0.937 \pm 0.002$ | $0.574 \pm 0.066$ | 776.6 |
| EQL-Z$_{w/oSTL}$ | $0.996 \pm 0.012$ | $0.898 \pm 0.261$ | $0.683 \pm 0.340$ | 70.4 |
| EQL-Z$_{w/oZP}$ | $0.995 \pm 0.018$ | $0.956 \pm 0.132$ | $0.838 \pm 0.379$ | 16.7 |
| EQL-Z | $\mathbf{1.000 \pm 0.000}$ | $\mathbf{1.000 \pm 0.000}$ | $\mathbf{1.000 \pm 0.000}$ | 17.5 |

*Table 6.* Complete 20-run component-wise exact recovery rates on the 12 standard Nguyen benchmarks. "w/o ZP", "w/o STL", and "w/o BFGS" denote removing the zero-point constraint, the small-to-large structure search, and the BFGS refinement step from the full EQL-Z pipeline, respectively. The last four columns isolate single-component variants built from the vanilla EQL backbone.

| Benchmark | EQL-Z | w/o ZP | w/o STL | w/o BFGS | STL only | BFGS only | ZP only | EQL | ParFam[†] |
|---|---|---|---|---|---|---|---|---|---|
| Nguyen-1 | 100% | 100% | 60% | 100% | 100% | 20% | 60% | 0% | 100% |
| Nguyen-2 | 100% | 100% | 30% | 100% | 100% | 10% | 10% | 0% | 100% |
| Nguyen-3 | 100% | 95% | 10% | 100% | 90% | 0% | 5% | 0% | 100% |
| Nguyen-4 | 100% | 90% | 0% | 100% | 80% | 0% | 0% | 0% | 100% |
| Nguyen-5 | 80% | 60% | 0% | 25% | 0% | 0% | 0% | 0% | 83% |
| Nguyen-6 | 100% | 100% | 0% | 100% | 65% | 0% | 0% | 0% | 83% |
| Nguyen-7 | 100% | 0% | 0% | 100% | 0% | 0% | 0% | 0% | 100% |
| Nguyen-8 | 100% | 100% | 100% | 100% | 100% | 100% | 100% | 0% | 100% |
| Nguyen-9 | 100% | 95% | 0% | 100% | 80% | 0% | 0% | 0% | 100% |
| Nguyen-10 | 100% | 100% | 0% | 75% | 70% | 0% | 0% | 0% | 100% |
| Nguyen-11 | 100% | 100% | 0% | 30% | 5% | 0% | 0% | 0% | 0% |
| Nguyen-12 | 100% | 90% | 15% | 100% | 45% | 0% | 0% | 0% | 100% |
| Average | 98.3% | 86.3% | 17.9% | 77.5% | 61.3% | 10.8% | 14.6% | 0.0% | 88.8% |

These observations highlight that structural design and operator-layer composition are as crucial as the operator library itself for successful equation recovery. Zero-point constraints and scaled initialization provide stability at the gradient and scale level, while the small-to-large strategy gradually filters out task-mismatched structures in architecture space. The combination of these components enables EQL-Z to find more compact analytic structures that closely match the ground truth, substantially improving equation recovery rates.

# F. Additional Plots and Tables

Figure 9 compares EQL-Z and baseline methods on ground-truth benchmarks, showing that EQL-Z achieves strong accuracy with markedly lower expression complexity; compared to the original EQL, it attains both higher accuracy and substantially reduced complexity. Table 6 reports the complete component-wise 20-run ablation on the Nguyen recovery benchmark. Table 2 reports ID/OOD NMSE on real-world tasks; EQL-Z consistently improves over the original EQL and is competitive with other non-LLM symbolic regression baselines, while remaining below LLM-SR methods that use large language model priors.

Table 6 further disentangles the contribution of each component. Among the single-component variants, STL only achieves the highest average recovery rate (61.3%), confirming that controlling structural expansion is the dominant factor for exact recovery in this benchmark. Nevertheless, the comparison between ZP only and vanilla EQL shows that the zero-point constraint is also effective in isolation: with the same backbone and without small-to-large search or BFGS refinement, adding zero-point-consistent operators improves the average recovery rate from 0.0% to 14.6%. This indicates that zero-point consistency is not merely a secondary effect of structure search, but directly changes the optimization behavior of EQL by reducing persistent interference from redundant operator paths. Finally, BFGS only improves recovery to 10.8%, suggesting that numerical refinement alone is insufficient when the symbolic structure is not already close to the target. The strongest results are obtained when all three components are combined, indicating that STL, ZP, and BFGS act at different stages of the recovery pipeline.

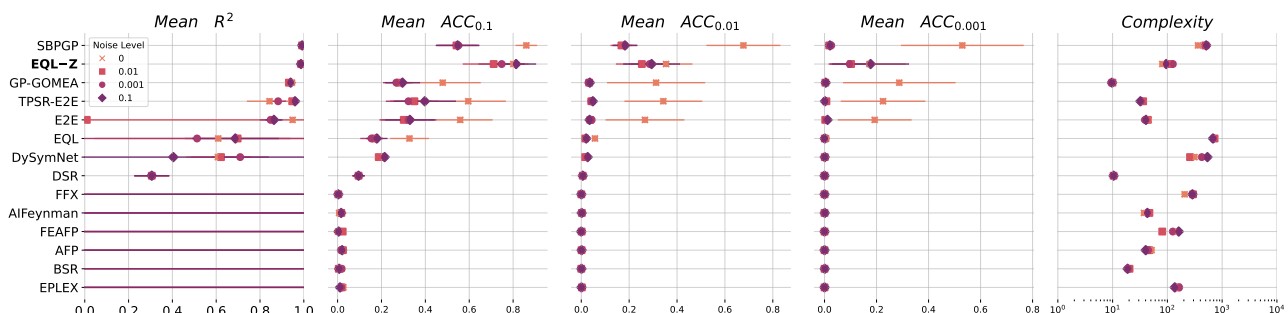

*Figure 9.* Performance and complexity comparison of EQL-Z and baseline methods on ground-truth datasets.

*Table 7.* Feature description of the MaiMemo memory dataset.

| Feature Name | Description |
| --- | --- |
| Word Difficulty $d$ | Intrinsic difficulty of the current word |
| Previous Review Interval $\Delta t$ | Time interval since the previous review of the word |
| Total Interval Sum $\sum \Delta t$ | Cumulative sum of all past review intervals |
| Total Review Count $n$ | Total number of times the word has been reviewed |
| User Proficiency $u$ | User's current proficiency on the word |
| Last Recall Success $s$ | Indicator of success in the last review ($s = 1$ if successful) |
| Weighted Success Rate $\hat{r}_w$ | Time-decayed weighted recall success rate |
| Overall Success Rate $\hat{r}$ | Fraction of successful recalls over the entire review history |
| Predicted Probability $\hat{y}$ | Predicted probability of successful recall in the current review (target) |

## G. Memory Modeling Experiments

To further assess the adaptability of EQL-Z on real-world tasks, we conduct a recall-probability prediction experiment on the MaiMemo memory dataset. This dataset is collected from the word-learning platform MoMo (MaiMemo) (Ye et al., 2022), which aims to provide efficient vocabulary learning and review strategies to improve users' memorization outcomes. During learning, MaiMemo records users' review behaviors, including word difficulty, learning trajectories, review intervals, recall outcomes, and the system's predicted recall probabilities. Compared with standard benchmarks such as PMLB, MaiMemo is much closer to real memory-modeling scenarios and is well suited for exploring the mathematical structure underlying users' cognitive behavior.

To ensure data quality, we retain only those samples with at least 100 observations under the same conditions, thereby filtering out unstable cases that may arise from noise or extreme individual differences. We then perform systematic feature engineering, including outlier removal (e.g., discarding samples with $\hat{y} = 0$ or $\hat{y} = 1$), variable transformations, and construction of statistical features. The core modeling variables, summarized in Table 7, cover key factors such as review intervals, user proficiency, and recall success rates, which jointly characterize the essential aspects of memory behavior.

We compare EQL-Z against several baseline models, including the Half-Life Regression (HLR) model (Settles & Meeder, 2016), the Wickelgren memory model (Wickelgren, 1974), an end-to-end regression model (E2E), and a tree-based symbolic regression model (TPSR). HLR assumes that memory decays exponentially over time, with the following functional form:

$$\hat{y} = \exp\left(-\frac{\ln 2 \cdot t}{h}\right),$$

where $h$ denotes the memory half-life, i.e., the time it takes for the recall probability to decay to $0.5$. The Wickelgren model instead assumes a power-law decay:

$$\hat{y} = \lambda \cdot t^{-\psi},$$

where $\lambda$ and $\psi$ are learnable parameters. We use mean squared error (MSE), root mean squared error (RMSE), and mean absolute error (MAE) as evaluation metrics. The experimental results are summarized in Table 8.

From the results, we observe that EQL-Z outperforms traditional memory models and existing symbolic regression methods

*Table 8.* Performance and learned formulas of different models on the MaiMemo dataset.

| Model | RMSE | MAE | MSE | Predicted Formula |
|-------|------|-----|-----|-------------------|
| EQL-Z | **0.07127** | **0.05736** | **0.0051** | $\hat{y} = 0.457 \cdot \exp(0.124x_3 - 0.079x_4) + 0.412$ |
| HLR | 0.2128 | 0.1618 | 0.0453 | $\hat{y} = \dfrac{x_2}{2^{-0.0380x_0 + 2.017x_1 + 0.3162x_3} + 0.0418x_4 - 0.4145x_5 + 0.6889}$ |
| Wick | 0.1267 | 0.0999 | 0.0160 | $\hat{y} = 0.7713 \cdot (1 + 1.7040x_1)^{0.0255}$ |
| E2E | 0.0898 | 0.0664 | 0.0081 | $\hat{y} = 0.98 - 0.00854 \cdot \sqrt{\lvert ABC \rvert} + 0.00667,$ $A = 0.00078x_3 - 6.037x_6 + 100.872,$ $B = 0.732x_5 - 5.888x_7 + 3.624,$ $C = 0.0325x_1 - 0.0217x_2 + 0.3565x_4$ $\quad - 3.0 \cdot \cos(102.67x_0 - 220.97) + 0.456$ |
| TPSR | 0.6090 | 0.5957 | 0.3709 | $\hat{y} = \dfrac{0.9}{0.1 + 8.99 - 0.001/(0.015 + 0.051) - 5.0/x_3}$ |

on all three metrics (RMSE, MAE, and MSE). This indicates that EQL-Z is better able to capture individualized memory patterns while maintaining a high degree of structural compactness.

