# OpenReview forum: "Stabilizing Equation Learning via Zero-Point Constraints"
_ICML.cc/2026/Conference — ICML 2026 regular_

### Official Review · Reviewer_TvyJ · 2026-03-08

**Soundness:** 3
**Presentation:** 3
**Significance:** 3
**Originality:** 3
**Overall Recommendation:** 4
**Confidence:** 2

**Summary:**

This paper addresses the problem of symbolic regression using neural networks, specifically focusing on the Equation Learner (EQL) framework. When symbolic operators do not vanish at zero (e.g., exp(0) = 1), they can induce non-zero gradients even when their coefficients are zero, causing redundant operators to be spuriously activated during training.

To address this issue, the authors propose \textbf{EQL-Z}, a modified framework that introduces a \textbf{zero-point constraint}, requiring symbolic operators to satisfy $\Lambda(0)=0$. This is implemented through simple operator transformations such as output shifts (e.g., $f(x) - f(0)$). The method also incorporates a \textbf{small-to-large architecture search strategy} that incrementally expands the symbolic network while controlling complexity. Finally, a \textbf{BFGS fine-tuning step} is applied to refine numerical coefficients after the symbolic structure has been determined.

Experiments on synthetic symbolic regression benchmarks (e.g., Nguyen functions), SRBench datasets, and several scientific tasks show that the proposed method improves equation recovery and produces more compact symbolic expressions compared with the original EQL.

Overall, the authors address a relevant challenge in neural symbolic regression: improving the stability and structural correctness of gradient-based symbolic models.

**Compliance With Llm Reviewing Policy:**

Affirmed.

**Final Justification:**

hank you for the detailed rebuttal and additional clarifications. I view the paper positively overall: the authors address a relevant challenge in neural symbolic regression, and the zero-point constraint is a simple yet meaningful idea that improves stability and equation recovery over vanilla EQL.

The rebuttal helped clarify several aspects of the experimental setup, runtime, and the role of different components, and I appreciate the additional experiments. However, consistent with concerns also raised by other reviewers, I still find that the evaluation could be stronger in terms of broader and more direct comparisons with recent EQL-style methods under a unified protocol, and the mechanistic claim remains only partially supported empirically.

For these reasons, while I remain supportive of the paper and believe it has merit, I will keep my score unchanged.

**Key Questions For Authors:**

1. Transforming operators changes what expressions the network can represent directly. For instance, if the true function is exp(x), the network with exp(x)−1 needs to learn an additional +1 bias term. This is mentioned briefly but not analyzed systematically. Does this cause any issues on benchmarks where the ground truth contains exp, cos, or other transformed operators? Have you observed any cases where this causes recovery failures? A systematic analysis would be helpful.
This can address comments on soundness of the model and some concerns on lack of theoretical depth.
2. Computational cost: How does the runtime of EQL-Z compare to vanilla EQL and to baselines like ParFam or PySR? How many structure candidates are typically evaluated during the small-to-large search?
3. Hyperparameter sensitivity: How sensitive are the results to the choice of $\delta$score, $\epsilon$layer, and other search hyperparameters? Were these tuned on a validation set, or are the reported values used consistently across all experiments?

**Limitations:**

Yes, the authors discuss limitations briefly in the Impact Statement. They acknowledge that EQL-Z does not yet match LLM-SR on real-world tasks. However, the limitations section could be more explicit about:
- When EQL-Z is expected to work well vs. poorly
- The computational overhead of the search procedure
- Scalability to larger input dimensions or more complex operator libraries

**Strengths And Weaknesses:**

Strengths
- Clear identification of a training instability in EQL. The paper provides a clear analysis explaining why redundant operators can remain active during training. The observation that operators with non-zero output at zero input can produce persistent gradients is intuitive and well motivated. The theoretical explanation that such paths prevent the ground-truth expression from being a stationary point is useful and highlights a limitation of existing EQL formulations.
- The proposed zero-point constraint is simple and easy to implement. Transforming operators so that $\Lambda(0)=0$ is a straightforward idea that could potentially apply to other symbolic neural networks as well. I appreciate that the solution does not introduce heavy architectural changes and instead modifies operator definitions.
- The experiments show clear improvements in equation recovery and structural compactness compared with the original EQL. For example, the method achieves a much higher recovery rate on the Nguyen benchmarks, while also producing simpler symbolic expressions. The ablation study also helps show the effect of both the zero-point constraint and the incremental structure search.
- The experimental section is organized around explicit research questions (RQ1–RQ5), which makes the evaluation easier to follow and helps clarify what aspects of the method are being tested.

Weaknesses
- Theoretical analysis remains somewhat limited.
The paper includes a theorem showing that redundant paths are not stationary points when $\Lambda(0) \neq 0$. However, the analysis focuses on a simplified setting and does not provide guarantees about:
    - convergence properties of the modified model,
    - recovery guarantees, or
    - generalization behavior.
The theory therefore explains a possible mechanism behind the observed instability, but it does not provide strong formal guarantees for the proposed method.
- Computational cost and search behavior are unclear.  This makes it difficult to understand the practical efficiency of the approach. The incremental small-to-large search procedure is an important part of the method, but the paper does not provide enough information about:

    - runtime cost,

     - number of structures evaluated,

     - sensitivity to hyperparameters.

- Real-world generalization (Table 5). EQL-Z improves substantially over EQL but still lags behind LLM-SR and sometimes PySR/uDSR. This is acknowledged, but it would help to discuss why and what would be needed to close this gap.
- Comparison scope. The main comparisons to vanilla EQL show large improvements, but EQL is known to have these instability issues. Comparisons to more recent variants like DySymNet or CONSOLE would strengthen the evaluation. The paper mentions these but does not compare directly in the main tables.

**Soundness.**
The theoretical analysis (Theorem 4.2) is correct and clearly stated. The proof in Appendix B is straightforward. The experimental methodology is reasonable: multiple random seeds, multiple initialization schemes, and ablations. The only concern is that the claim that zero-point constraints "guarantee" silent paths (Section 4.1.1) holds only when weights AND inputs are zero. The paper could discuss more carefully when this condition holds during training.

**Presentation.**
The paper is generally well-written and organized. The problem is well-motivated, and the figures (especially Figure 1 showing gradient evolution) are helpful. Some areas for improvement:
 - Section 4.2 is dense. The algorithm description would benefit from a cleaner presentation or a worked example.
 - The paper uses "small-to-large" terminology but sometimes calls it "incremental structure search." Consistent terminology would help.
  - Some symbols are introduced without clear definition (e.g., Cmax in the score function).
  - "trainable constants and linear coefficients" in Section 4.3 could be clarified.

**Significance.**
The paper makes a useful contribution to neural symbolic regression. The zero-point constraint is a simple idea that addresses a real problem. The improvements over vanilla EQL are substantial. The gap to LLM-SR suggests that incorporating language model priors may be more impactful for real-world tasks.

**Originality.**
The combination of zero-point constraints with incremental search is novel for symbolic regression. The theoretical observation about gradient residuals is a nice contribution. The originality is solid but not exceptional.

---

> ### Author Rebuttal · Authors · 2026-03-31
>
> Thank you for the careful reading and positive evaluation. We greatly appreciate your recognition of our analysis of EQL training instability, the simplicity of the zero-point constraint, and the organization of the experiments around different research questions. The issues you raised are all very important, and we will address them one by one in the revision.
>
> **1. The transformed operators do not fundamentally reduce expressivity, but they do introduce additional offset terms, which should be simplified when exporting the final formula.**
> Our zero-point-consistent transformations are essentially output shifts or input shifts of the original operators. Therefore, they do not systematically reduce the class of functions that the model can represent, but they can change the “minimal parameter form” of certain expressions. For example, if the target function contains $\exp(x)$, the original form can be written as $a\exp(bx)+c$; for the transformed operator $\mathrm{exp\_reg}(x)=\exp(x)-1$, the corresponding form becomes $a(\exp(bx)-1)+c$, where the ideal constant term is shifted accordingly. As a result, exported formulas may contain expressions such as $\exp(x)-1+1$, which are semantically equivalent but include extra offset terms. In the revision, we will clarify more explicitly that zero-point transformations do not change reachability of the target function, but they can introduce offset terms that should be further simplified / beautified in the final exported expression.
>
> **2. Small-to-large does introduce additional search cost, and we will report runtime and search behavior more clearly in the revision.**
> With 15-way parallelism, the runtime of EQL-Z is typically about 10 minutes to 6 hours, depending on task difficulty and the search process, and the small-to-large search evaluates about 50 candidate structures on average. As a reference, the ParFam paper reports a runtime scale of about 27 hours, although it does not specify the degree of parallelism; PySR typically takes about 30 minutes to 8 hours under 25-way parallelism. In the revision, we will provide a clearer summary of runtime, parallel settings, and the average number of evaluated candidate structures.
>
> **3. The main experiments on the Nguyen benchmarks use a unified hyperparameter configuration, and we are also adding a more systematic hyperparameter analysis.**
> For the main Nguyen results, we use a unified hyperparameter setting rather than tuning separately for each task. Specifically, the EQL-family experiments use a unified training setup with learning rate 0.01, $L_1=0.01$, and 10,000 training epochs. In the structure-search stage, we use $L_{\text{start}}=1$, $L_{\max}=2$, max\_ops\_per\_layer=3, operator\_level\_start=0, and operator\_level\_max=5, and initialize the search from seed structures such as `[['id']]`, `[['id','square']]`, `[['id','square','cube']]`, and `[['sin']]`. The main experiments use a 1200s time budget, run on an RTX 4090, and use 15 parallel jobs. We agree that a systematic hyperparameter sensitivity analysis is important. We are currently adding this part, but due to the limited rebuttal time, it has not yet been fully completed; in the revision, we will provide a clearer description of the parameter settings and include sensitivity results for key hyperparameters as much as possible.
>
> **4. We will make the limitations more explicit in the revision, especially for the real-world results.**
> We will state more clearly that, although EQL-Z substantially improves over vanilla EQL on real-world tasks, it still lags behind LLM-SR. We believe an important reason is that LLM-SR benefits from stronger structure-search priors provided by large models. We will make this point more explicit in the revision and frame it as a promising future direction: combining structure priors from large models with the continuous optimization ability of EQL to further improve structure search on complex tasks. For higher-dimensional inputs or more complex operator libraries, both the search cost and the overall modeling difficulty can increase substantially.  Regarding EQL-style variants such as DySymNet, CONSOLE, and iEQL, please see Point 4 of our response to Reviewer 418g for a more detailed clarification of comparison scope and experimental positioning.
>
> **5. We also greatly appreciate the writing and presentation issues you pointed out, and we will correct them one by one in the revision.**
> This includes wording that may currently be too strong (such as “guarantee silent paths”), inconsistent terminology between “small-to-large” and “incremental structure search,” unclear symbol definitions (e.g., $C_{\max}$), and imprecise phrasing such as “trainable constants and linear coefficients.” We will revise and unify these points in the revision, and we will also try to add a more intuitive algorithm description or a worked example to make Section 4.2 easier to follow.

---

> > ### Author Rebuttal · Reviewer_TvyJ · 2026-04-02
> >
> > Thank you for the helpful rebuttal. It clarified the experimental setup, runtime, and the intended scope of the method more clearly. However, my main concern remains the limited comparison with other methods beyond ParFam. Overall, I think the current score is fair, so I will keep it unchanged.

---

> > > ### Author Response · Authors · 2026-04-08
> > >
> > > Thank you for the reviewer’s further feedback. We agree that the presentation of comparisons with related methods beyond ParFam could be made clearer in the current version.
> > >
> > > First, we would like to clarify that our paper does not compare EQL-Z only against ParFam. In fact, the main paper already includes more than ten representative symbolic regression baselines spanning multiple major methodological paradigms, including the state-of-the-art LLM-based method LLM-SR [1], the classical method PySR [2], end-to-end generation-based methods such as E2E [3] and TPSR [4], as well as network-like methods such as ParFam [5] and DySymNet [6]. Therefore, our evaluation of EQL-Z is not built on comparison with a single baseline, but on a systematic comparison against multiple representative methods under a unified experimental protocol.
> > >
> > > In addition, to further address the reviewer’s concern about related-method comparisons, during the rebuttal stage we additionally evaluated two reproducible EQL-related baselines, namely eEQL and OccamNet. Under the same experimental setting as used in the main paper, EQL-Z shows better overall stability and practical usability. Specifically, OccamNet exhibited noticeable instability in our supplementary experiments, including extremely negative $R^2$, abnormally inflated errors, and even `nan/inf` on some datasets. In contrast, the main issue with eEQL was not simple numerical divergence, but rather its high search cost under a fixed budget: it often remained at intermediate expressions that were already highly complex while still delivering unsatisfactory predictive performance. By comparison, EQL-Z achieved more stable fitting results and more compact expressions under the same budget. See reviewer 418g for details.
> > >
> > > Therefore, the advantage of EQL-Z is not limited to a pointwise comparison against ParFam, but is also reflected in systematic comparisons with multiple representative methods, where it demonstrates better stability, more compact expressions, and stronger practical usability.
> > >
> > > [1] Shojaee, P., Meidani, K., Gupta, S., Farimani, A. B., & Reddy, C. K. *LLM-SR: Scientific Equation Discovery via Programming with Large Language Models*. In *The Thirteenth International Conference on Learning Representations*.
> > >
> > > [2] Cranmer, M. (2023). *Interpretable machine learning for science with PySR and SymbolicRegression.jl*. arXiv preprint arXiv:2305.01582.
> > >
> > > [3] Kamienny, P. A., d'Ascoli, S., Lample, G., & Charton, F. (2022). *End-to-end symbolic regression with transformers*. In *Advances in Neural Information Processing Systems*, 35, 10269-10281.
> > >
> > > [4] Shojaee, P., Meidani, K., Barati Farimani, A., & Reddy, C. (2023). *Transformer-based planning for symbolic regression*. In *Advances in Neural Information Processing Systems*, 36, 45907-45919.
> > >
> > > [5] Scholl, P., Bieker, K., Hauger, H., & Kutyniok, G. (2025). *ParFam--(Neural Guided) Symbolic Regression via Continuous Global Optimization*. In *The Thirteenth International Conference on Learning Representations*.
> > >
> > > [6] Li, W., Li, W., Yu, L., Wu, M., Sun, L., Liu, J., ... & Hao, M. (2024). *A Neural-Guided Dynamic Symbolic Network for Exploring Mathematical Expressions from Data*. In *International Conference on Machine Learning* (pp. 28222-28242). PMLR.

---

### Official Review · Reviewer_XqaR · 2026-03-10

**Soundness:** 2
**Presentation:** 3
**Significance:** 2
**Originality:** 2
**Overall Recommendation:** 4
**Confidence:** 3

**Summary:**

This paper revisits Equation Learner (EQL) for symbolic regression and argues that a key source of its unstable structure recovery is a gradient residual effect caused by operators that do not vanish at zero.
The authors show, through toy examples and gradient analysis, that even when the target expression is representable and the model is initialized near the ground-truth structure, redundant operator paths can still be spuriously activated, so the ideal sparse solution may fail to be a local optimum.
Based on this analysis, the paper proposes EQL-Z, an EQL-based framework that enforces zero-point constraints through zero-point-consistent operator transformations, uses an incremental small-to-large structure search with complexity-aware validation to grow compact architectures, and optionally applies BFGS coefficient refinement after structure selection.
The paper evaluates the method on Nguyen benchmarks, SRBench black-box tasks, scientific discovery datasets, and a real-world memory dataset, reporting much higher exact equation recovery and much lower expression complexity than vanilla EQL, while remaining competitive with strong non-LLM symbolic regression baselines.
Overall, the paper’s main contribution is a training-mechanism perspective on why EQL over-activates redundant structures, together with a practical framework for improving symbolic recovery stability and compactness.

**Compliance With Llm Reviewing Policy:**

Affirmed.

**Final Justification:**

I recommend weak accept. The paper studies an important problem and proposes a reasonably original combination of zero-point constraints, small-to-large search, and coefficient refinement to improve EQL-based symbolic regression. In my evaluation, the main strengths are the potential significance of the problem, the practical usefulness of the method, and the improved empirical results. My main concerns were about soundness and attribution, particularly whether the claimed mechanism was established strongly enough and whether the respective roles of STL, ZP, and BFGS were sufficiently disentangled. The rebuttal did not fully remove these concerns, and I still think there are some limitations in baseline fairness and in how strongly the mechanistic claim is supported. However, it did improve the clarity and calibration of the paper, and it provided enough additional evidence to make the contribution more convincing overall. On balance, while the work is not without weaknesses, I believe it is sufficiently interesting, useful, and supported to justify a weak accept recommendation.

**Key Questions For Authors:**

1. Can the authors provide stronger evidence that the proposed “nonzero-at-zero operator” gradient-residual effect is a primary cause of EQL’s practical failure modes, rather than just one possible failure mechanism?

The current support seems to rely mainly on a toy case and a small set of controlled gradient visualizations, while the paper makes a broader causal claim about EQL instability. A stronger empirical diagnosis across more targets/operator sets would increase my confidence in the central mechanistic claim.

2. What is the precise relationship between the zero-point argument and the small-to-large search strategy?

In the current presentation, the zero-point constraint is motivated by the gradient analysis, but the small-to-large component reads more like a general structure-search heuristic based on complexity-penalized validation than a direct consequence of that analysis. Clarifying whether small-to-large is theoretically motivated by the same mechanism, or mainly an effective heuristic, would help me better assess the paper’s conceptual contribution.

3. Can the authors strengthen the recovery-rate evidence with more runs and a fairer baseline protocol?

The paper reports EQL-family recovery rates over only six random seeds, while several competing baselines are taken directly from ParFam rather than rerun under the same experimental setup. Since exact recovery is a discrete and potentially high-variance metric, additional runs, confidence intervals, or same-protocol reruns would materially affect how convincing I find the empirical results.

4. How much of the final improvement is attributable to zero-point constraints versus small-to-large search versus BFGS refinement?

The paper argues that combining zero-point-consistent operators with incremental structure search is key, but the current evidence still leaves some ambiguity about which component is doing most of the work, especially for exact recovery. A clearer component-wise attribution could improve my assessment of both soundness and originality.

**Limitations:**

No. The paper would benefit from a more explicit discussion of limitations. In particular, the central mechanistic claim is supported mainly by a toy-task analysis and controlled gradient examples, which may be insufficient to justify the broader claim that this is the main cause of EQL’s failure in practice.

The method also depends on several heuristic design choices beyond the zero-point idea itself, including user-specified or heuristically chosen initial structures and an incremental small-to-large search procedure.

Finally, the exact-recovery evaluation would benefit from a clearer discussion of statistical uncertainty and baseline comparability, since recovery is a discrete metric and the protocol combines repeated runs for EQL-Z with baseline numbers drawn from prior work.

**Strengths And Weaknesses:**

**Soundness**

The paper has a reasonable core intuition and the technical story is coherent at a local level: the authors analyze a setting in which operators with $\Lambda(0)\neq 0$ can induce nonzero gradients on redundant paths, and they propose zero-point-consistent operator transformations to suppress such activations.   However, I am not fully convinced by the causal attribution. The analysis appears to establish a relatively narrow non-stationarity mechanism, but the paper does not sufficiently show that this is the main reason for EQL’s practical failure modes. In addition, the exact-recovery evidence is not very strong statistically: for EQL-family methods, recovery rates are averaged over only six runs, and other baselines are taken directly from the ParFam paper rather than being rerun under the same protocol. Since recovery rate is a high-variance discrete metric, this limits how strongly I can trust the reported differences.

**Presentation**

The paper is generally readable and the progression from observation to analysis to method is easy to follow. The motivation for zero-point constraints is clearly stated, and the paper is explicit that EQL-Z combines zero-point-consistent operators, small-to-large structure search, and BFGS refinement.   My main presentation issue is that the link between the analysis and the final method is overstated. In particular, the small-to-large search is presented as part of the same training-mechanism story, but in the paper it reads more like a separate search heuristic based on validation score and complexity penalty than a direct consequence of the gradient-residual analysis.

**Significance**

The problem is relevant: improving structure recovery in differentiable symbolic regression is worthwhile, and the paper does show clear gains over vanilla EQL in recovery rate and compactness.  That said, I view the contribution as somewhat narrow. The paper is mainly an improvement to the EQL family rather than a broader advance for symbolic regression, and the current empirical evidence is not strong enough for me to conclude that the proposed mechanism and solution will generalize beyond this setting.

**Originality**

The paper contains a useful idea, namely connecting EQL instability to zero-point inconsistency of operators and designing transformed operators that satisfy $\Lambda(0)=0$.   I do think this is more than a trivial engineering tweak. However, the overall method is less original than the framing suggests, because an important part of the final gain seems to come from the incremental small-to-large structure search, which is only loosely tied to the main theoretical argument and functions more as an architecture-search heuristic.

---

> ### Author Rebuttal · Authors · 2026-03-31
>
> Thank you for the careful reading and constructive comments. We fully agree with your concerns regarding the strength of the causal attribution, the relationship among components, and the statistical sufficiency of the evidence.
>
> **1. We agree that the current version overstates the mechanistic claim, and in the revision we will more accurately frame it as “an important but not unique failure mechanism.”**
> Your point is well taken. In the revision, we will soften phrases such as “main/primary cause” and instead emphasize that zero-point inconsistency is an *important and practically relevant mechanism*. More specifically, it is not the only cause, but once it is activated, it creates a structural obstacle to exact recovery and therefore cannot be ignored.
>
> **2. We agree that the relationship between the zero-point argument and the small-to-large strategy should be clarified more carefully, and in the revision we will describe them as a “mechanism-driven intervention” and a “complementary search strategy,” respectively.**
> More precisely, zero-point consistency is an operator-level intervention directly motivated by the gradient-residual analysis: it explains why redundant paths can be spuriously activated. By contrast, small-to-large is not a direct theoretical consequence of that analysis, but rather a *complementary complexity-aware search strategy*. At the same time, it is not an arbitrary heuristic unrelated to the zero-point idea: the two share the same practical goal of reducing redundant structures. In the revised version, we will provide two related theoretical results to help clarify this design motivation: first, removing redundant operators reduces parameter count and structural complexity; second, under the condition $\Lambda(0)=0$, removing redundant operators does not disturb the gradients of the remaining parts and helps simplify the optimization space. Based on this redundancy-reduction principle, the core idea of small-to-large is that, rather than starting from an overly large network with many potentially redundant operators and relying on pruning afterward, it is preferable to begin with a compact structure and expand only when needed. In this sense, the former mainly addresses *operator-level residual activation*, while the latter mainly addresses *structure-level search difficulty*.
>
> **3. We have expanded the Nguyen recovery experiments from 6 to 20 runs, and the new results provide clearer support for both the statistical claims and the component attribution.**
> The new 20-run results are reported in **Table 1** of our response to **Reviewer rcAR**. The full EQL-Z model achieves an average recovery rate of **98.3%**, compared with **86.3%** without the zero-point constraint, **17.9%** without small-to-large, and **77.5%** without BFGS. Comparing **EQL vs. EQL+zp** and **EQL+BFGS vs. EQL+BFGS+zp** shows that simply introducing zero-point-consistent operator transformations already leads to a clear improvement. Likewise, `EQL_{w_zp}=14.6%` versus `EQL_v=0.0%` indicates that EQL’s failure is not merely due to generic optimization difficulty. For several formulas, merely changing the operator form (i.e., introducing zero-point-consistent operators) already changes recovery from 0 to non-zero, which suggests that zero-point inconsistency is itself a real operator-level limitation. Furthermore, comparing **EQL+stl+BFGS** with the full **EQL-Z** also reveals the additional contribution of Z. We would also like to emphasize that BFGS itself is a standard post-hoc refinement step and is not the main novelty of our paper; rather, our focus is on what the zero-point constraint and small-to-large contribute beyond a standard refinement step. At the same time, because small-to-large is itself designed to reduce redundant operators, part of the gain that Z would otherwise show in isolation can be partially masked once STL is introduced; in other words, STL already alleviates part of the redundancy problem at the structural level, so on some benchmarks the marginal improvement from Z becomes smaller than in the no-STL setting. **At the same time, small-to-large remains crucial for recovery, while BFGS mainly performs coefficient refinement on already promising symbolic candidates.** **Overall, the gains of the full method come from the complementary roles of Z, small-to-large, and BFGS, rather than from any single component alone.**  We will also make these limitations explicit in the revised manuscript.

---

> > ### Author Rebuttal · Reviewer_XqaR · 2026-04-03
> >
> > Thank you for the detailed rebuttal. I appreciate that the authors clarified several points and, in particular, moderated the original mechanistic claim by reframing the nonzero-at-zero operator gradient-residual effect as an important but not unique failure mechanism. I also find the additional 20-run recovery-rate results helpful, since they strengthen the component-wise empirical picture and make it clearer that the final performance is not attributable solely to a post-hoc BFGS refinement step.
> >
> > That said, these clarifications are not sufficient to change my overall assessment. My main concern was whether the paper had convincingly established this mechanism as a central explanation for EQL’s practical failure modes across settings, and I do not think the rebuttal fully resolves that issue. In my reading, the rebuttal mainly weakens the causal claim to a more defensible one, rather than substantially strengthening the underlying mechanistic evidence. Similarly, while the relationship between the zero-point argument and the small-to-large strategy is now described more clearly, the rebuttal also makes clear that small-to-large is better viewed as a complementary search strategy than as a direct theoretical consequence of the gradient analysis. This improves conceptual clarity, but it does not materially strengthen the paper’s unified theoretical contribution.
> >
> > On the empirical side, the expanded recovery experiments are useful, but I still have some reservations about evaluation rigor and attribution. Exact recovery is a high-variance discrete metric, and while increasing from 6 to 20 runs is a meaningful improvement, my concern about baseline fairness is only partially addressed, since some comparisons still appear not to be rerun under the same protocol. More broadly, although the new ablations suggest that zero-point constraints, small-to-large search, and BFGS play complementary roles, I do not think the rebuttal fully settles the question of which component is most responsible for the reported gains, especially from the standpoint of methodological originality.
> >
> > For these reasons, I appreciate the authors’ clarifications and do think the rebuttal improves the presentation and calibration of the claims, but it does not sufficiently change my view of the paper’s soundness and originality to justify raising my score.

---

> > > ### Author Response · Authors · 2026-04-08
> > >
> > > Thank you for the reviewer’s detailed feedback. After revisiting this issue, we believe our response mainly concerns the following four points.
> > >
> > > **1. On whether our theoretical claim was overstated, and whether the relation between the theory and the strategy was described inaccurately**
> > >
> > > Our previous wording was not precise enough. The theory identifies an important, though not exclusive, failure mode: redundant operators can continuously interfere with optimization through the residual-at-zero effect. ZP mitigates this at the operator level, while STL mitigates it at the structure-search level. Thus, both address the same underlying issue from different levels.
> > >
> > > **2. On the role of STL, ZP, and BFGS respectively, and why they are complementary**
> > >
> > > These three components operate at different levels. STL is the most direct and effective one, because it reduces the chance that interfering terms enter candidate structures and simultaneously shrinks the search space, making sparse optimization in EQL easier. By contrast, ZP does not reduce the search space; instead, it weakens the persistent optimization interference caused by redundant terms within a given structure, and thus improves optimization tolerance when such terms are present. This also explains why STL and ZP work better together: STL reduces the occurrence of interfering terms, while ZP further mitigates their effect if they still remain. BFGS serves a different purpose, as a post hoc numerical refinement step that pushes an already nearly correct expression toward strict exact recovery.
> > >
> > > **3. On whether the current experiments really support the above mechanism explanation and component attribution**
> > >
> > > We believe the current experiments support this explanation from several angles.
> > >
> > > First, in the simplified setting of Experiment 2, even when the target expression itself is very simple, standard gradient descent may still fail once only a few redundant operators are added. This suggests that the issue is not merely large-scale search complexity; rather, redundant terms themselves can continuously interfere with optimization.
> > >
> > > Second, Nguyen-1 and Nguyen-8 provide more direct evidence in the full search setting. Under the same full-operator search space, vanilla EQL can achieve high $R^2$ yet still converge to local optima containing multiple redundant terms rather than the true exact expression. For Nguyen-1, the obtained result is
> > > $-0.37\exp(\min(-0.32,\ 0.60-0.29x_0))+1.52\exp(\min(1.13,\ 1.64x_0-0.66))+0.55/|x_0|^{0.01}+0.95\sqrt{|0.12x_0+0.11|}-1.38|0.36x_0+0.46|^{\min(0.58,\ |0.67x_0+0.87|)}-0.01\sqrt{|1.42x_0-0.64x_1+0.25|}-0.51$
> > >
> > > and for Nguyen-8, the obtained result is
> > > $0.01\exp(\min(-0.01,\ 0.004-0.002x_0))-0.01\exp(\min(-0.003,\ 0.01x_0+0.01))+0.90\sqrt{|x_0|}+0.23\sqrt{|0.19x_0|}+0.003$.
> > > These examples suggest that the issue is not insufficient fitting, but that the optimization trajectory is diverted by redundant terms. Under the same setting, adding ZP enables exact recovery, indicating that its role is not merely to improve fitting accuracy, but to change the optimization behavior in the presence of interfering terms.
> > >
> > > Finally, the successful STL cases show a consistent pattern: the final recovered structures exclude the interfering terms discussed in the theory. For example, Nguyen-3 and Nguyen-4 both yield `[["cube", "id", "square"], ["mul", "id"]]`, while Nguyen-5 yields `[["square", "id"], ["sin", "sin"]]`, corresponding to $0.5\sin(x^2-x)+0.5\sin(x^2+x)-1$, which is equivalent to the target $\sin(x^2)\cos(x)-1$. This suggests that STL helps not only by shrinking the search space, but also by favoring structures that exclude operators likely to cause persistent optimization interference.
> > >
> > > **4. On whether the overall experimental trend and comparison setting support our conclusion**
> > >
> > > The overall ablation trend supports a complementary interpretation of STL, ZP, and BFGS. Across 20 repeated runs on the 12 Nguyen tasks, the full EQL-Z achieves 98.3\% average exact recovery. Among single-component variants, STL is clearly the strongest (61.3\%), outperforming ZP-only (14.6\%) and BFGS-only (10.8\%). At the same time, adding ZP on top of STL further improves recovery from 61.3\% to 77.5\%, while ZP+BFGS without STL reaches only 17.9\%. This suggests that STL provides the main structural benefit, while ZP offers additional robustness once interfering terms remain, and BFGS mainly helps final numerical refinement.
> > >
> > > As for computational budget, our setting is of the same order as ParFam: ParFam uses large-scale parallel candidate evaluation with total runtime around 27 hours, while we use a 24-hour budget with smaller but still substantial parallelism. We therefore believe the comparison is at least broadly fair in terms of compute.

---

### Official Review · Reviewer_418g · 2026-03-11

**Soundness:** 2
**Presentation:** 3
**Significance:** 3
**Originality:** 3
**Overall Recommendation:** 4
**Confidence:** 3

**Summary:**

The paper introduces an augmentation of the EQL framework denoted as EQL-Z. EQL-Z introduces two novel alterations to the EQL framework. 1) EQL-Z makes use of a zero point constraint, $\Lambda(0) = 0$, to ensure that any path with 0 weights produces a 0 as an output. 2) EQL-Z enables the network to dynamically deepen or widen to increase the richness of the network if the richness results in a measurable improvement. The paper evaluates EQL-Z on three popular benchmarks, highlighting its improvements over EQL and its comparability with other symbolic regression methods. An ablation study is conducted by removing either component to quantify the effect of each on EQL-Z's performance.

**Compliance With Llm Reviewing Policy:**

Affirmed.

**Final Justification:**

The reviewer weakly recommends acceptance of this paper. The rebuttal improved the reviewer's confidence in their understanding of the results and clarified EQL-Z's scope within Symbolic Regression. EQL-Z provides a novel implementation of the EQL framework, but it relies heavily on comparisons with EQL to justify its improvements rather than other SOTA EQL-based methods. The primary EQL-based comparison for EQL-Z is ParFam, which did not result in a significant difference in performance. The reviewer weakly recommends acceptance of this paper due to its potential to improve existing EQL-based methods and relative performance with Symbolic Regression, but notes a lack of rigor in the comparison with them.

**Key Questions For Authors:**

1. Could the authors provide more information about the set up for the experiments? The reviewer is confused as to the difference in Figure 9 compared to results reported in the original SRBench paper.
2. Could the authors provide runtimes for their method and comparable methods?
3. Could the authors provide hyperparameters for their method?
4. The reviewer would be interested in understanding how EQL-Z fits into the current EQL landscape. Does EQL-Z offer an alternative to other SOTA EQL methods or can the augmentations proposed apply to other SOTA EQL methods?

The reviewer finds the paper to be technically sound, but finds oversights in the paper's experimentation. The reviewer would support increasing their score should these oversights, three weaknesses listed above, be adequately addressed or acknowledged.

**Limitations:**

Yes

**Strengths And Weaknesses:**

Strengths:
- New methodology: EQL-Z provides two notable augmentations to the standard EQL framework to improve performance.
- Interpretability: EQL-Z provides a significant reduction in complexity compared to EQL, enabling a high degree of interpretability of the discovered expression.
- Ablations: The paper provides ablations of the components, enabling experimental justification for the augmentations to EQL.

Weaknesses:
- Limited comparable methods: The paper validates performance on a range of benchmarks, but has limited comparison to alternative EQL methods. While the paper highlights EQL-Z improvement over EQL, it does not provide a comparison to SOTA EQL methods.
- Runtimes: The paper does not acknowledge any runtimes or computation requirements.
- Reproducibility: The paper does not provide sufficient details about the experiments to enable accurate reproduction. Missing details include trial counts, computation or time budget, resources, and hyperparameters.

---

> ### Author Rebuttal · Authors · 2026-03-31
>
> Thank you for the careful and constructive review. We are glad that you found the paper technically sound, and we agree that the current version does not yet provide sufficient detail on experimental setup, runtime, and the positioning of EQL-Z within the broader EQL landscape. We will clarify these points in the rebuttal and revision.
>
> **1.We will provide a more complete description of the experimental setup and reproducibility details.**
> For the EQL-family experiments, we use a unified training configuration with learning rate 0.01, $L_1=0.01$, and 10,000 training epochs. In the structure-search stage, we use $L_{\text{start}}=1$, $L_{\max}=2$, max_ops_per_layer=3, operator_level_start=0, and operator_level_max=5, and initialize the search from several seed structures, including [['id']], [['id','square']], [['id','square','cube']], and [['sin']]. The main experiments use a 1200s time budget, run on an RTX 4090, and use 15 parallel jobs. In addition, we are recomputing the recovery-related results under 20 independent runs to reduce sensitivity to random seeds and single-run fluctuations.
>
> **2. The difference between Figure 9 and the original SRBench results mainly comes from the evaluation protocol and time budget.**
> In the SRBench part, we did not aim to reproduce the full long-horizon search protocol used in the original paper; instead, we used a 10-minute time budget to evaluate EQL-Z under a constrained-budget setting. As stated in the paper, the SRBench black-box tasks are used to assess performance under a tight computation budget. Since the structure search in our method is mainly based on heuristic expansion and selection, the overall computational cost is still relatively high. Excluding the SRBench setting, the average wall-clock runtime of the other experiments (e.g., equation recovery experiments) is roughly between 10 minutes and 6 hours (with 15 parallel jobs). We will include a clearer summary of time budgets and resource configurations in the revision.
>
> **3. We will make the position of EQL-Z within the current EQL landscape more explicit.**
> From a methodological perspective, EQL-Z can be viewed both as a complete EQL variant and as a combination of two relatively orthogonal enhancements: zero-point constraint, which mitigates residual gradients on redundant paths in end-to-end gradient-based training, and small-to-large structure search, which controls structural expansion in large search spaces. More broadly, our results suggest that under the joint effect of small-to-large structural control and zero-point constraint, neural-like end-to-end symbolic models can indeed achieve meaningful equation recovery ability. The paper already evaluates this from multiple perspectives, including Nguyen exact recovery, SRBench black-box / ground-truth tasks, ID/OOD generalization on scientific problems, and a real-world memory modeling task. We also believe that future work can build on this framework with more efficient structure-search strategies.
>
> **4.We will also clarify the current scope of comparison with newer related methods.**
> A substantial part of symbolic regression research focuses on exact recovery, but existing EQL-style methods are often not strong on this metric; many of them focus more on fitting error, search efficiency, or structural reachability than on stable formula recovery. Against this background, we paid particular attention to ParFam, DysymNet, CONSOLE, and iEQL. Among them, CONSOLE and iEQL do not currently provide publicly available code that we could directly reproduce, so we have not yet been able to conduct a sufficiently fair unified-protocol comparison. For DysymNet, exact recovery is not the main evaluation target in the original paper, and in our own reproduction its recovery performance was not strong, so we did not include it in the main recovery-rate table. As for ParFam, although its authors do not strictly position it as an EQL-family method, it is still an important and strong related baseline, so we keep it in our comparisons. Therefore, while we do not yet have sufficiently broad coverage of newer EQL-family baselines, we do compare against several very strong recent symbolic regression baselines, including LLM-SR, PySR, and TPSR, so that EQL-Z is also positioned against competitive contemporary SR methods more broadly.

---

> > ### Author Rebuttal · Reviewer_418g · 2026-04-02
> >
> > The reviewer thanks the authors for their responses to all of the reviewers. The reviewer will maintain their score, as they find the responses insufficient for improvement. The reviewer's primary concern is the lack of comparison between EQL-Z and existing EQL methods [1, 2, 3, 4] beyond ParFam in Figure 1. While the reviewer recognizes that EQL-Z can improve existing EQL frameworks, the reviewer finds the lack of comparison with EQL methods insufficient to justify a higher score. The reviewer finds the score fair, as EQL-Z relies exclusively on Table 1 to compare EQL-Z to ParFam.
> >
> > [1] Jiří Kubalík, Erik Derner, Robert Babuška (2023, Feb). Toward Physically Plausible Data-Driven Models: A Novel Neural Network Approach to Symbolic Regression.
> >
> > [2] Sahoo, S., Lampert, C., & Martius, G. (2018, July). Learning equations for extrapolation and control. In International Conference on Machine Learning (pp. 4442-4450). Pmlr.
> >
> > [3] Junlan Dong, Jinghui Zhong, Wei-Li Liu , and Jun Zhang (2025, October). Evolving Equation Learner for Symbolic Regression.
> >
> > [4] A. Costa et al., “Fast neural models for symbolic regression at scale,”
> > 2020, arXiv:2007.10784.

---

> > > ### Author Response · Authors · 2026-04-08
> > >
> > > We appreciate the reviewer’s concern that the comparison with EQL-family methods is not yet sufficiently comprehensive, and we would like to further clarify the position of EQL-Z within this line of work.
> > >
> > > First, the original EQL proposed by Sahoo et al. (2018) [2] is already included in our current experimental comparison. Since EQL-Z is a direct improvement built upon the original EQL framework, this baseline is also the most direct and most important reference for evaluating whether our method is effective.
> > >
> > > For the other methods mentioned by the reviewer, we further examined their reproducibility and compatibility with our experimental protocol during the rebuttal stage. Our paper follows the SRBench black-box symbolic regression setting, with a unified 10-minute time budget for each dataset. For [1], we agree that it is a relevant neuro-symbolic regression method. However, during the rebuttal period, we were unable to identify a publicly available implementation that could be directly reproduced and aligned fairly with our protocol. Therefore, we are currently unable to provide a comparison that we consider sufficiently fair and reliable.
> > >
> > > That said, we additionally evaluated two reproducible EQL-related baselines, namely eEQL [3] and OccamNet [4]. The supplementary results are shown below. We report $R^2$ (mean ± std), ACC01 (mean ± std), and formula complexity $C_1$ (mean) under different noise levels.
> > >
> > > |Method|Noise|R2(mean±std)|ACC01(mean±std)|C1(mean)|
> > > |---|---:|---:|---:|---:|
> > > |EQL-Z|0.000|0.82±0.31|0.40±0.29|88.18|
> > > |EQL-Z|0.001|0.76±0.40|0.49±0.34|42.89|
> > > |EQL-Z|0.010|0.84±0.25|0.40±0.28|72.62|
> > > |EQL-Z|0.100|0.83±0.26|0.38±0.23|68.13|
> > > |DySymNet|0.000|0.75±0.34|0.40±0.30|199.39|
> > > |DySymNet|0.001|0.77±0.33|0.33±0.26|332.09|
> > > |DySymNet|0.010|0.70±0.35|0.33±0.29|203.36|
> > > |DySymNet|0.100|0.53±1.71|0.34±0.26|411.06|
> > > |OccamNet|0.000|-0.31±1.30|0.01±0.01|10.74|
> > > |OccamNet|0.001|-0.32±1.34|0.01±0.02|13.91|
> > > |OccamNet|0.010|-0.30±1.27|0.01±0.01|11.70|
> > > |OccamNet|0.100|-0.34±1.48|0.01±0.01|11.09|
> > > |eEQL|0.000|-643028.27±3267725.14|0.01±0.02|7415.85|
> > > |eEQL|0.001|-17300.68±86710.14|0.01±0.02|7131.15|
> > > |eEQL|0.010|-508033.43±2340170.24|0.01±0.01|7348.74|
> > > |eEQL|0.100|-4510.51±15241.93|0.003±0.01|7332.74|
> > >
> > > From these results, we observe that under the unified SRBench black-box symbolic regression setting and the same 10-minute budget, EQL-Z shows better overall stability and practical usability.
> > >
> > > For OccamNet, our supplementary experiments revealed clear instability, including extremely negative $R^2$, abnormally large MSE/MAE values, and even `nan/inf` on some datasets. For example, on `1028_SWD`, OccamNet produced extremely negative values together with `inf`; on `1193_BNG_lowbwt`, its MSE/MAE values were also evidently abnormal. Even after excluding these clearly unstable cases, the average $R^2$ over the remaining datasets is still negative, suggesting that under the unified black-box evaluation protocol used in this paper, OccamNet does not achieve satisfactory overall fitting performance. We understand that this may be related to its stronger reliance on large-scale sampling and parallel exploration, whose advantages are harder to realize under a strict and unified time budget.
> > >
> > > By contrast, the issue with eEQL does not primarily appear to be simple numerical divergence, but rather insufficient search progress under a limited budget. Within the fixed time budget, it often remains stuck at intermediate expressions that are already highly complex but still perform poorly. On tasks such as `feynman_I_9_18` and `feynman_II_36_38`, for example, the candidate expressions obtained within the same 10-minute budget are typically already very complicated, while their predictive performance remains unsatisfactory. The table reflects the same pattern: across all noise levels, eEQL yields much lower average $R^2$, while its mean formula complexity stays around 7.1k--7.4k, which is substantially higher than that of EQL-Z. This is also consistent with the characteristics of eEQL, which relies on ADF and progressive structural evolution. When the search budget is strictly limited, its structural search cannot be sufficiently expanded, which in turn affects both stability and compactness.
> > >
> > > DySymNet can achieve reasonable average $R^2$ under some settings, but its formula complexity is consistently much higher than that of EQL-Z, and its variance becomes much larger under higher noise. For example, at noise = 0.1, its $R^2$ standard deviation reaches 1.71, while the mean complexity increases to 411.06. In contrast, EQL-Z maintains relatively stable $R^2$ and ACC01 across different noise levels, while keeping the formula complexity within a much more reasonable range. This indicates that the advantage of EQL-Z lies not only in fitting quality, but also in stability and expression usability.

---

### Official Review · Reviewer_rcAR · 2026-03-12

**Soundness:** 3
**Presentation:** 3
**Significance:** 3
**Originality:** 3
**Overall Recommendation:** 4
**Confidence:** 4

**Summary:**

This paper finds the unstable training cause of the EQL model and propose a simple but effective method to fix this problem. The unstable training derived from the specific operators whose gradient is not zero when the input is zero. This introducing abundant terms for non-zero gradient even training from the right structure. This method solves the long-standing problem of unstable training that has plagued the EQL algorithm.

**Compliance With Llm Reviewing Policy:**

Affirmed.

**Final Justification:**

The authors have satisfactorily addressed all of my concerns. This paper effectively tackles a critical issue in EQL-based methods, prompting me to raise my score to 4.

**Key Questions For Authors:**

1. The paper lacks specialized handling for the division operator and reciprocal functions, which are undefined when the denominator approaches zero. Notably, the tested benchmark expressions do not include the division operator.

2. In Section 5.4, using only six independent runs per benchmark is relatively limited. For evaluating recovery rates, experiments should be conducted with at least 20 independent runs.

3. The ablation study in Table 1 suggests that the small-to-large strategy contributes significantly to search progress. However, with only six independent runs, the improvement on benchmarks such as nguyen5 and nguyen5_c is marginal—only one additional successful recovery of the ground-truth expression.

4. To better illustrate the problem addressed—specifically, that the ideal expression is not a local optimum—the authors could include more challenging expressions that require operator transformations to demonstrate meaningful progress.

**Limitations:**

1. The paper avoids the division operation, but division operation is necessary.

2. There are issues with the experimental parameter settings for the recovery rate, and the insufficient number of runs does not provide sufficient evidence to demonstrate the effectiveness of the method.

**Strengths And Weaknesses:**

"Strengths:"
This paper focuses on an inherent issue of the EQL method: even when initialized with the correct structure, it tends to evolve toward an incorrect solution by introducing irrelevant nodes. The work identifies the underlying cause of this phenomenon and proposes a simple yet effective approach to address it. Additionally, the introduced small-to-large learning strategy demonstrates strong effectiveness.

"Weaknesses:"
While this paper addresses certain problems of EQL, fundamental issues in its structure and updating mechanism remain unresolved. For example, the division operator easily leads to numerical overflow, and heterogeneous activation functions cause scaling discrepancies between nodes, which can negatively affect training. Moreover, the experimental validation in the paper is limited to a narrow set of simple benchmarks.

---

> ### Author Rebuttal · Authors · 2026-03-31
>
> Thank you for the valuable comments. We greatly appreciate your concerns regarding the statistical strength of the recovery-rate evaluation and the coverage of division, and we have further organized our rebuttal and revision plan accordingly.
>
> **1. Our empirical evaluation is not limited to simple benchmarks, but already covers multiple levels of tasks.**
> Specifically, the paper evaluates EQL-Z on exact recovery over the Nguyen benchmarks, black-box and ground-truth tasks from SRBench, ID/OOD generalization on four scientific discovery problems, and a real-world memory modeling task. Thus, our experiments are intended to assess EQL-Z from multiple perspectives, including structure recovery, generalization, and applicability to real tasks.
>
> **2. We have expanded the Nguyen recovery experiments to 20 independent runs, and the updated results are shown in Table 1.**
> The 20-run results confirm the main conclusion under a stronger statistical protocol: EQL-Z still substantially outperforms vanilla EQL, and the updated results further show that **both the zero-point constraint and the structural search strategy are important**. On some simpler formulas, one component mainly provides an incremental gain; on more challenging formulas, **both the zero-point constraint and the small-to-large search are needed** for stable recovery.
>
> Table 1
> |Benchmark|EQL-Z| EQL-Z$_{+stl+bfgs}$ | EQL-Z$_{+zp+bfgs}$ | EQL-Z$_{+zp+stl}$ | EQL$_{+stl}$ | EQL$_{+bfgs}$ | EQL$_{+zp}$ | EQL$_{v}$ | ParFam$^\dagger$ |
> |-----------|------:|-------------------:|--------------------:|---------------------:|---------------:|----------------:|--------------:|----------:|-----------------:|
> |Nguyen-1|100%|100%|60%|100%|100%|20%|60%|0%|100%|
> |Nguyen-2|100%|100%|30%|100%|100%|10%|10%|0%|100%|
> |Nguyen-3|100%|95%|10%|100%|90%|0%|5%|0%|100%|
> |Nguyen-4|100%|90%|0%|100%|80%|0%|0%|0%|100%|
> |Nguyen-5|80%|60%|0%|25%|0%|0%|0%|0%|83%|
> |Nguyen-6|100%|100%|0%|100%|65%|0%|0%|0%|83%|
> |Nguyen-7|100%|0%|0%|100%|0%|0%|0%|0%|100%|
> |Nguyen-8|100%|100%|100%|100%|100%|100%|100%|0%|100%|
> |Nguyen-9|100%|95%|0%|100%|80%|0%|0%|0%|100%|
> |Nguyen-10|100%|100%|0%|75%|70%|0%|0%|0%|100%|
> |Nguyen-11|100%|100%|0%|30%|5%|0%|0%|0%|0%|
> |Nguyen-12|100%|90%|15%|100%|45%|0%|0%|0%|100%|
> |Average|98.3%|86.3%|17.9%|77.5%|61.3%|10.8%|14.6%|0.0%|88.8%|
>
> **3. We focus on one key redundant-activation mechanism in EQL, and we will revise any wording that may overstate the scope of this claim.**
> When some operators satisfy \(\Lambda(0)\neq 0\), redundant paths that should remain silent can still receive non-zero gradients, making the ideal sparse expression fail to be a stable local optimum. The zero-point constraint is introduced precisely to cut off such residual gradients and alleviate redundancy in EQL.
>
> **4. Division can also be incorporated into our framework.**
> Prior EQL-family work (e.g., EQL÷[1]) has already introduced protected/regularized division, for example
>
> $$
> h^\theta(a,b)=
> \begin{cases}
> \dfrac{a}{b}, & \text{if } b>\theta,\\
> 0, & \text{otherwise.}
> \end{cases}
> $$
>
> which explicitly suppresses output and gradient in the small-denominator region so that the corresponding path becomes “silent.” This is conceptually aligned with our analysis of silent-path stability and residual-gradient elimination, rather than being in conflict with it. What we have not yet systematically covered is a unified zero-point-consistent treatment of raw division and related singular operators, and we will clarify this boundary in the revision.
>
> [1] Sahoo, S., Lampert, C., & Martius, G. (2018, July). Learning equations for extrapolation and control. In International conference on machine learning (pp. 4442-4450). Pmlr.

---

> > ### Author Rebuttal · Reviewer_rcAR · 2026-04-04
> >
> > Thank you for the detailed rebuttal and the additional 20-run experiments. However, my core concern regarding the operator library's completeness—specifically the exclusion of the division operator—remains unaddressed.
> >
> > While the theoretical compatibility of "protected division" is noted, SR in real-world applications operates without prior knowledge of the target expression. Consequently, a robust SR algorithm must search within a comprehensive baseline operator library.
> >
> > Excluding division from the benchmark experiments (e.g., Nguyen) fundamentally weakens the evaluation for two critical reasons:
> >
> > - **Search space complexity:** Introducing additional operators exponentially expands the search space, making structural discovery significantly harder.
> >
> > - **Gradient vulnerability:** The EQL framework relies heavily on gradient propagation. Even when "protected," division introduces severe numerical singularities that frequently disrupt gradient-based optimization and induce redundant activations.
> >
> > Validating EQL-Z solely within a customized, smooth operator sub-space avoids the method's most critical stress test. To convincingly demonstrate that EQL-Z genuinely resolves the "redundant activation" issue, it is imperative to evaluate the exact equation recovery rate on the Nguyen dataset using an expanded, comprehensive operator library (explicitly including division).
> >
> > Without empirical evidence proving that EQL-Z can stably converge to the ground truth amidst the optimization interference of singular operators, the method's practical robustness and real-world utility remain unverified.

---

> > > ### Author Response · Authors · 2026-04-08
> > >
> > > Thank you for the reviewer’s further comments. We understand the concern regarding the completeness of the operator library. We would like to clarify that the additional 20 runs were not conducted in a customized search space containing only smooth operators. Instead, they already used the full operator library, including division:
> > > `['id', 'square', 'cube', 'sin', 'cos_reg', 'mul', 'add', 'sqrt_reg', 'exp_reg', 'log_reg', 'pow_reg', 'div_reg']`.
> > >
> > > In other words, these experiments were performed in a unified search space that already included division, as well as other operators that are more challenging for optimization, rather than pre-selecting operators based on the target expression.
> > >
> > > Under this setting, we observed that although introducing division increases the search difficulty, it does not eliminate the recovery ability of EQL-Z. For example, on Nguyen-1 and Nguyen-2, without using any prior knowledge and directly searching over the full operator library (including division), EQL+zp was still able to recover the correct expression to a certain extent. This suggests that the zero-point constraint is not only effective in simplified operator spaces; even in a more difficult setting with division, it can still help mitigate the optimization interference caused by division and suppress redundant activations. Meanwhile, in our actual training, we also applied gradient clipping to control gradient magnitude, so protected division did not lead to direct gradient explosion in our experiments.
> > >
> > > In addition, we further tested two Feynman formulas from PMLB, both of which contain division:
> > >
> > > - Feynman-I-18-4: $y=\frac{m_1 r_1 + m_2 r_2}{m_1 + m_2}$
> > > - Feynman-I-39-22: $y=\frac{T k_b n}{V}$
> > >
> > > Among them, EQL-Z achieved 100\% exact recovery on Feynman-I-18-4. This result indicates that EQL-Z not only alleviates the optimization difficulty introduced by division, but can also recover the true expression with division on some tasks. For Feynman-I-39-22, the challenge comes not only from division itself, but also from the multiplicative structure in the numerator. Although this formula is still representable by the current operator set in principle, the present architecture is not particularly well suited for such expressions, making the search more difficult. In future work, we will consider incorporating richer operator structures to improve recoverability on this type of formula.

---

### Decision · Program_Chairs · 2026-04-30

**Decision:**

Accept (regular)

**Comment:**

This paper addresses unstable training in the Equation Learner (EQL) framework for symbolic regression. The authors identify that operators which do not output zero when given a zero input (e.g., $exp(0)=1$) create "residual gradients" that prevent the network from successfully pruning redundant paths. To fix this, they introduce EQL-Z, a framework utilizing zero-point consistent operator transformations, an incremental "small-to-large" architecture search, and BFGS fine-tuning.

Strengths
- Identifies a specific, intuitive mathematical flaw causing optimization failures in standard EQL.
- The zero-point constraint is a simple fix.
- Empirical gains in both exact equation recovery and model compactness compared to vanilla EQL.

Weaknesses
- The initial submission lacked statistical confidence
- Lack of comparison against more modern, state-of-the-art EQL variants.

Rebuttal
- The authors increased the number of seeds to 20
- The authors added comparisons with eEQL and OccamNet

While the contribution is more practical than a major breakthrough, the empirical improvements to the EQL framework seem significant. The paper successfully diagnoses a known optimization bottleneck in symbolic regression and provides an effective solution.

Suggestion by the AC: the title feels a bit awkward. I would recommend a more concise title such as "Stabilizing Equation Learning via Zero-Point Constraints".